# Meiosis-specific ZFP541 repressor complex promotes developmental progression of meiotic prophase towards completion during mouse spermatogenesis

Yuki Horisawa-Takada [1], Chisato Kodera[1,2,11], Kazumasa Takemoto [1,11], Akihiko Sakashita[3], Kenichi Horisawa[4], Ryo Maeda[5], Ryuki Shimada [1], Shingo Usuki[6], Sayoko Fujimura[6], Naoki Tani[6], Kumi Matsuura[7], Tomohiko Akiyama[8], Atsushi Suzuki [4], Hitoshi Niwa[7], Makoto Tachibana[5], Takashi Ohba[2], Hidetaka Katabuchi[2], Satoshi H. Namekawa[9], Kimi Araki [10] & Kei-Ichiro Ishiguro [1]✉

During spermatogenesis, meiosis is accompanied by a robust alteration in gene expression and chromatin status. However, it remains elusive how the meiotic transcriptional program is established to ensure completion of meiotic prophase. Here, we identify a protein complex that consists of germ-cell-specific zinc-finger protein ZFP541 and its interactor KCTD19 as the key transcriptional regulators in mouse meiotic prophase progression. Our genetic study shows that ZFP541 and KCTD19 are co-expressed from pachytene onward and play an essential role in the completion of the meiotic prophase program in the testis. Furthermore, our ChIP-seq and transcriptome analyses identify that ZFP541 binds to and suppresses a broad range of genes whose function is associated with biological processes of transcriptional regulation and covalent chromatin modification. The present study demonstrates that a germ-cell specific complex that contains ZFP541 and KCTD19 promotes the progression of meiotic prophase towards completion in male mice, and triggers the reconstruction of the transcriptional network and chromatin organization leading to post-meiotic development.

---

[1] Department of Chromosome Biology, Institute of Molecular Embryology and Genetics (IMEG), Kumamoto University, Kumamoto, Japan. [2] Department of Obstetrics and Gynecology, Faculty of Life Sciences, Kumamoto University, Kumamoto, Japan. [3] Department of Molecular Biology, Keio University School of Medicine, Tokyo, Japan. [4] Division of Organogenesis and Regeneration, Medical Institute of Bioregulation, Kyushu University, Fukuoka, Japan. [5] Graduate School of Frontier Biosciences, Osaka University, Osaka, Japan. [6] Liaison Laboratory Research Promotion Center, IMEG, Kumamoto University, Kumamoto, Japan. [7] Department of Pluripotent Stem Cell Biology, IMEG, Kumamoto University, Kumamoto, Japan. [8] Department of Systems Medicine, Keio University School of Medicine, Tokyo, Japan. [9] Department of Microbiology and Molecular Genetics, University of California, Davis, CA, USA. [10] Institute of Resource Development and Analysis, and Center for Metabolic Regulation of Healthy Aging, Kumamoto University, Kumamoto, Japan. [11]These authors contributed equally: Chisato Kodera, Kazumasa Takemoto. ✉email: ishiguro@kumamoto-u.ac.jp

Meiosis consists of one round of DNA replication followed by two rounds of chromosome segregation, producing haploid gametes from diploid cells. Meiotic entry is followed by meiotic prophase that corresponds to a prolonged G2 phase in which meiosis-specific chromosomal events such as chromosome axis formation, homolog synapsis, and meiotic recombination sequentially occur[1–3]. Completion of the meiotic prophase is regulated by sexually dimorphic mechanisms, so the transcription, cell cycle, and chromatin status are altered in the subsequent developmental program for sperm production and oocyte arrest/maturation. In male germ cells, the completion of pachytene is monitored under several layers of regulation such as pachytene checkpoint and meiotic sex chromosome inactivation (MSCI)[4–6]. Pachytene spermatocytes acquire competency for meiotic prophase–Metaphase I transition indicted by the response to phosphatase inhibitor okadaic acid (OA)[7], although the precise mechanism remains elusive. Male meiotic prophase is accompanied by robust changes of gene expression programs[8–12] and epigenetic status[13–16] as well as reorganization of the chromatin structure[17–19] for post-meiotic development. At the pachytene stage of male meiotic prophase, the transcriptional program for post-meiotic stages starts to take place[13,20,21]. A germline-specific Polycomb protein, SCML2 is, at least in part, responsible for the suppression of somatic/progenitor genes and the activation of late-spermatogenesis-specific genes in pachytene spermatocytes and round spermatids[22–24]. In addition to intrinsic gene expressions of the germ cells, extrinsic genes of the Sertoli cells that are responsive to androgen signaling make the germ cells permissive for the completion of the meiotic prophase and the subsequent first division phase[25]. Thus for spermatocytes, passing through the pachytene stage during meiotic prophase is a critical developmental event for the subsequent spermatid differentiation. However, it remained elusive how the completion of the meiotic prophase was ensured prior to post-meiotic differentiation.

In this study, we identify a protein complex that establishes the meiotic transcription program to ensure the completion of the meiotic prophase. Previously, we identified MEIOSIN that plays an essential role in meiotic initiation both in males and female[26]. MEIOSIN together with STRA8[27] acts as a crucial transcription factor that drives meiotic gene activation. In the present study, we identified the *Zfp541* gene that encodes a zinc-finger protein, as one of the MEIOSIN/STRA8-target genes. Although ZFP541 was previously identified as a zinc-finger protein SHIP1 in the spermatocyte UniGene library showing testis-specific expression[28], its function was yet to be determined.

Here we show that ZFP541 plays a role in promoting the developmental progression of meiotic prophase towards the completion in male germ cells. By mass spectrometry (MS) analysis, we show that ZFP541 is expressed at pachytene and interacts with KCTD19 and HDAC1/2. Disruption of *Zfp541* and *kctd19* leads to defects in the completion of the meiotic prophase, with a severe impact on male fertility. Chromatin binding analysis of ZFP541 combined with transcriptome analysis demonstrates that ZFP541 binds to and represses a broad range of genes whose biological functions are associated with the processes of transcriptional regulation and covalent chromatin modification. In round spermatids, those ZFP541-target genes are repressed with repressive histone marks. The present study suggests that ZFP541-containing complex globally represses the gene expression in the meiotic prophase, and triggers the reconstruction of transcription and chromatin organization to proceed with the developmental program for sperm production.

## Results

### ZFP541 is expressed during meiotic prophase in spermatocytes and oocytes.

Previously, we identified a germ-cell-specific transcription factor, MEIOSIN, that directs initiation of meiosis[26]. Our study demonstrated that MEIOSIN works with STRA8 and activates hundreds of meiotic genes, which are required for meiotic initiation. In spermatocytes, we identified *Zfp541* as one of the MEIOSIN/STRA8-bound genes during preleptotene that is the time of meiotic initiation (Fig. 1a). Expression of *Zfp541* was significantly downregulated in RNA-seq analysis of *Meiosin* KO testes at postnatal day 10 (P10)[26] which is the time when a cohort of spermatocytes undergo the first wave of meiotic entry. We confirmed this by RT-qPCR analysis demonstrating that *Zfp541* expression level was downregulated in *Meiosin* KO testis at P10 (Fig. 1b). However, *Zfp541* expression level was comparable between P10 WT and *Stra8* KO testes, consistent with our previous observation that the expression levels of some meiotic genes including *Zfp541* remained unaffected in *Stra8* KO, and the cytological phenotypes were milder in *Stra8* KO than in *Meiosin* KO[26]. We further examined the expression patterns of *Zfp541* in different mouse tissues by RT-PCR analysis. *Zfp541* gene was specifically expressed in adult testis, and weakly in embryonic testis, but not in other adult organs that we examined (Fig. 1c). Spermatogenic expression of the *Zfp541* gene was further confirmed by the reanalysis of previous scRNA-seq data of adult mouse testis[29] (Supplementary Fig. 1a). The result indicated that *Zfp541* was coordinately expressed with the landmark genes of meiosis such as *Spo11*, rather than those of spermiogenesis such as *Prm1*, and spermatogonia such as *Zbtb16*, along the pseudotime of spermatogenic development (Supplementary Fig. 1b). *Zfp541* expression was also detected in embryonic ovaries (Fig. 1d). These results suggest that *Zfp541* is a germ-cell-specific factor both in male and female, which is partly consistent with a previous study[28].

It has been suggested that ZFP541 possesses putative DNA-binding domains, C2H2 type zinc-finger, and ELM2-SANT domains[28]. However, its biological function has remained elusive. To determine the meiotic stage/cell type-specific expression of ZFP541, spread nuclei and seminiferous tubules of the WT testes (8-weeks old) were immunostained with specific antibodies against ZFP541 along with SYCP3 (a component of meiotic axial elements) (Fig. 1e, f). Immunostaining of the spread nuclei showed that ZFP541 appeared in the nuclei of the spermatocytes from pachytene onward, and in round spermatids (Fig. 1e). Consistently, the ZFP541 signal started to appear faintly in early pachytene spermatocyte nuclei of stage I seminiferous tubules, and persisted in the round spermatids of stage VII seminiferous tubules (Fig. 1f and Supplementary Fig. 2). However, it was not observed in spermatocytes before pachytene, elongated spermatids, or spermatogonia (Fig. 1f and Supplementary Fig. 2). Testis-specific histone H1t is a marker of spermatocytes later than mid pachytene[7,30]. Immunostaining of seminiferous tubules by testis-specific histone H1t indicated that localization of the ZFP541 protein into the nuclei started at the H1t-negative early pachytene stage (Fig. 1g), confirming the aforementioned observation. Although the expression of *Zfp541* mRNA was upregulated upon meiotic entry, immunostaining of ZFP541 protein detected no more than background levels in spermatocytes before the pachytene stage (Fig. 1e), suggesting that the expression of ZFP541 may be post-transcriptionally regulated after the entry into meiotic prophase I. In females, ZFP541 was detected at the protein level in the SYCP3-positive oocytes in the embryonic ovary during E13.5-18.5, the stage when oocytes progressed through the meiotic prophase (Supplementary Fig. 3a). This observation was confirmed by the reanalysis of previous scRNA-seq data of fetal ovaries[31]. We found that the gene expression of *Zfp541* was coordinately upregulated with that of *Stra8* along the pseudotime of fetal oocyte development (Supplementary Fig. 3b), although we had not specified yet whether MEIOSIN-STRA8

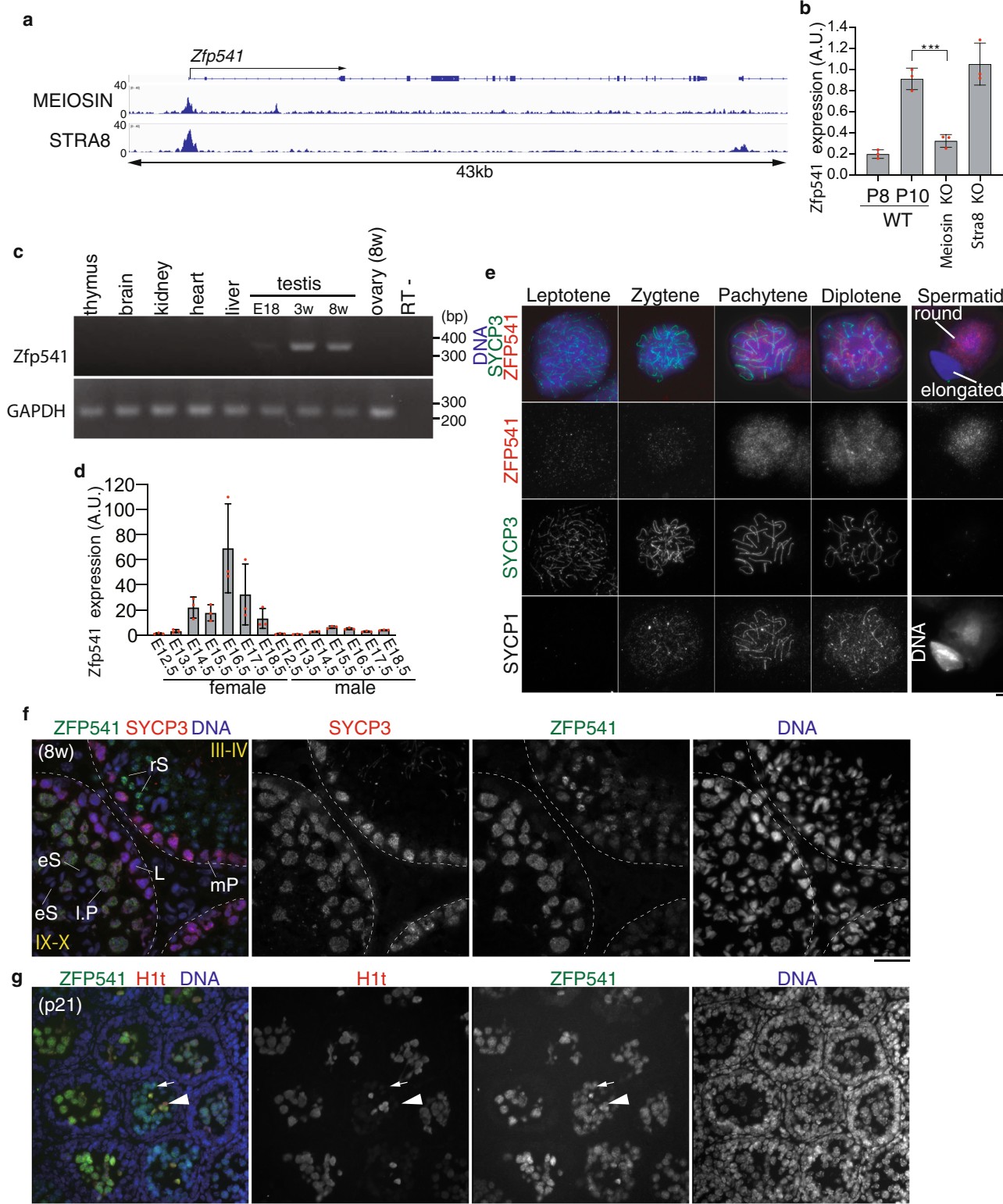

complex was involved in the ovarian *Zfp541* expression as in the spermatocytes. Altogether, it was shown that ZFP541 was expressed in meiotic prophase both in male and female.

***Zfp541* knockout in mice results in male infertility**. In order to address the role of ZFP541, we deleted Exon1–Exon13 of the *Zfp541* loci in C57BL/6 fertilized eggs by CRISPR/Cas9-mediated genome editing (Fig. 2a). Immunoblotting of the extract

from *Zfp541* KO testis showed that ZFP541 protein was absent (Fig. 2b), which was further confirmed by diminished immunolocalization of ZFP541 in the seminiferous tubules of *Zfp541* KO (Fig. 2f), indicating the targeted *Zfp541* allele was null. Although *Zfp541* KO mice developed normally, defects in male reproductive organs were evident with smaller-than-normal testes (Fig. 2c). Histological analysis revealed that post-meiotic spermatids and spermatozoa were absent in the

**Fig. 1 Zfp541 was identified as a MEIOSIN-STRA8 target gene. a** Genomic view of MEIOSIN and STRA8 binding peaks over the *Zfp541* locus. Genomic coordinates derived from Ensembl. **b** The expression of *Zfp541* in WT (P8 and P10), *Meiosin* KO, and *Stra8* KO testes was examined by RT-qPCR. Three animals for each genotype were used. The graph shows the expression level of *Zfp541* normalized by that of *GAPDH* with SD. The expression level of *Zfp541* in P10 WT was set to 1. Statistical significance (two-tailed *t*-test) is shown by ***p-value = 0.999 × 10⁻³. Biologically independent mice (N = 3) for each genotype were examined. **c** The tissue-specific expression pattern of *Zfp541* was examined by RT-PCR. Testis RNA was obtained from embryonic day 18 (E18), 3-weeks-old (3w), and 8-weeks-old (8w) male mice. Ovary RNA was obtained from adult 8-weeks-old (8w) female mice. RT− indicates control PCR without reverse transcription. The data were acquired from two separate experiments. **d** The expression patterns of *Zfp541* in the embryonic ovary and testis were examined by RT-qPCR. Average values normalized to E12.5 gonad are shown with SD from technical triplicates. N = 1 gonadal sample for each embryo. **e** Chromosome spreads of WT spermatocytes were immunostained as indicated. Biologically independent mice (N = 2) were examined in two separate experiments. Scale bar: 5 μm. **f** Seminiferous tubule sections in WT testis (8-weeks old) were immunostained as indicated. Lep: leptotene, Pa: pachytene, l.P: late pachytene spermatocyte, rS: round spermatid, eS: elongated spermatid. Boundaries of the seminiferous tubules are indicated by white dashed lines. Roman numbers indicate the seminiferous tubule stages. Biologically independent mice (N = 3) were examined in three separate experiments. Scale bar: 25 μm. **g** Seminiferous tubule sections in WT testis (P21) were immunostained as indicated. Arrow and arrowhead indicate H1t-negative and H1t-positive pachytene spermatocyte, respectively. Biologically independent mice (N = 5) were examined in two separate experiments. Scale bar: 25 μm.

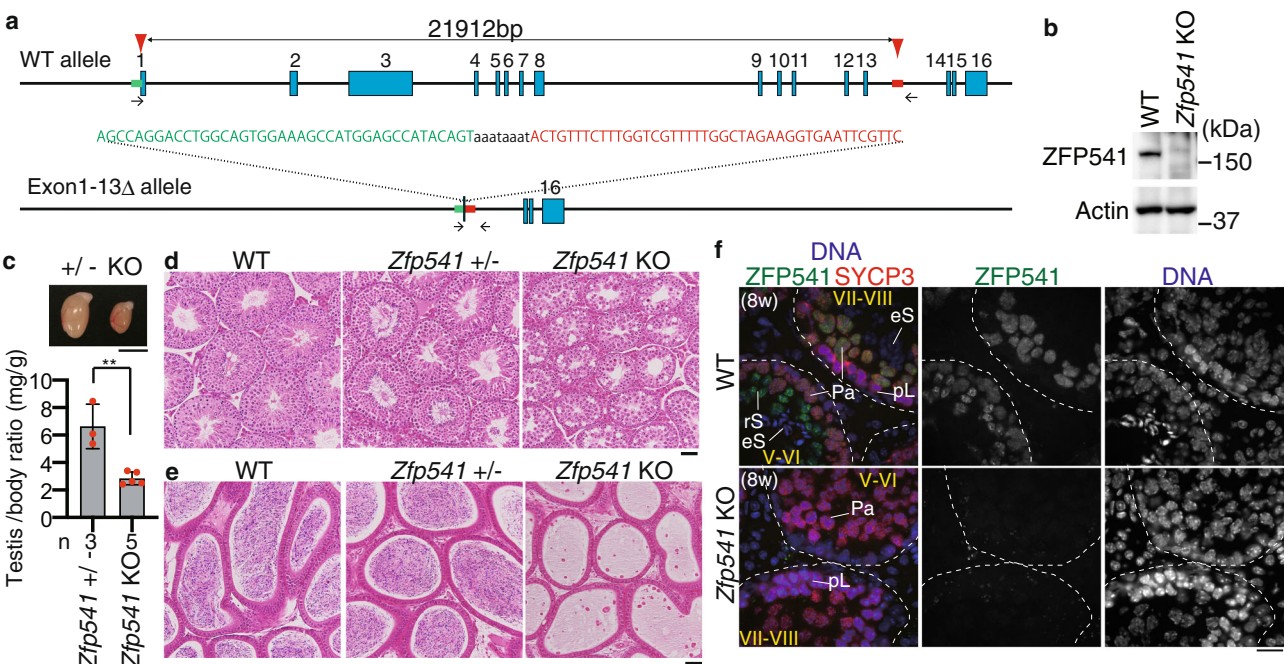

**Fig. 2 Spermatogenesis was impaired in Zfp541 knockout male. a** The allele with targeted deletion of Exon1–13 in *Zfp541* gene was generated by the introduction of CAS9, the synthetic gRNAs designed to target Exon1 and the downstream of Exon13 (arrowheads), and ssODN into C57BL/6 fertilized eggs. Arrows: PCR primers for genotyping. Four lines of KO mice were established. Line #16 of *Zfp541* KO mice was used in most of the experiments, unless otherwise stated. **b** Immunoblot analysis of testis extracts prepared from mice with the indicated genotypes (P21). Biologically independent mice (N = 2) were examined in two separate experiments. **c** Testes from *Zfp541*± and *Zfp541* KO (8-weeks old). Testis/body-weight ratio (mg/g) of *Zfp541*± and *Zfp541* KO mice (8-weeks old) is shown below (mean with SD). *n*: the number of animals examined. Statistical significance is shown by **p = 0.0021 (two-tailed *t*-test). Scale bar: 5 mm. **d** Hematoxylin and eosin staining of the sections from WT, *Zfp541*±, and *Zfp541* KO testes (8-weeks old). Biologically independent mice (N = 3) for each genotype were examined. Scale bar: 50 μm. **e** Hematoxylin and eosin staining of the sections from WT, *Zfp541*±, and *Zfp541* KO epididymis (8-weeks old). Biologically independent mice (N = 3) for each genotype were examined. Scale bar: 50 μm. **f** Seminiferous tubule sections (8-weeks old) were stained for SYCP3, ZFP541, and DAPI. pL: preleptotene, Pa: pachytene spermatocyte, rS: round spermatid, eS: elongated spermatid. Boundaries of the seminiferous tubules are indicated by white dashed lines. Roman numbers indicate the seminiferous tubule stages. Biologically independent mice (N = 3) for each genotype were examined. Scale bars: 15 μm.

*Zfp541* KO seminiferous tubules (Fig. 2d, f). Accordingly, sperm was absent in *Zfp541* KO caudal epididymis (Fig. 2e). Consistently, seminiferous tubules that contain PNA lectin (a marker of spermatids)-positive cells were absent in *Zfp541* KO (Supplementary Fig. 4a, b). Thus, the later stage of spermatogenesis was severely abolished in *Zfp541* KO seminiferous tubules, resulting in male infertility. In contrast to male, *Zfp541* KO females exhibited seemingly normal fertility with no apparent defects in adult ovaries (Supplementary Fig. 3c, d). Therefore, these results suggested that the requirement of ZFP541 is sexually different.

**ZFP541 is required for meiotic prophase completion in male.** To further investigate at which stage the primary defect appeared in the *Zfp541* KO, we analyzed the progression of spermatogenesis by immunostaining. Immunostaining analysis with antibodies against SYCP3 (a component of meiotic chromosome axis) along with SYCP1 (a marker of homolog synapsis) and γH2AX (a marker of DNA damage response for meiotic recombination and XY body), demonstrated that spermatocytes initiated double-strand breaks (DSBs) for meiotic recombination and underwent homologous chromosome synapsis in juvenile *Zfp541* KO males (P21) as in the age-matched control (Fig. 3a). However, spermatocytes later

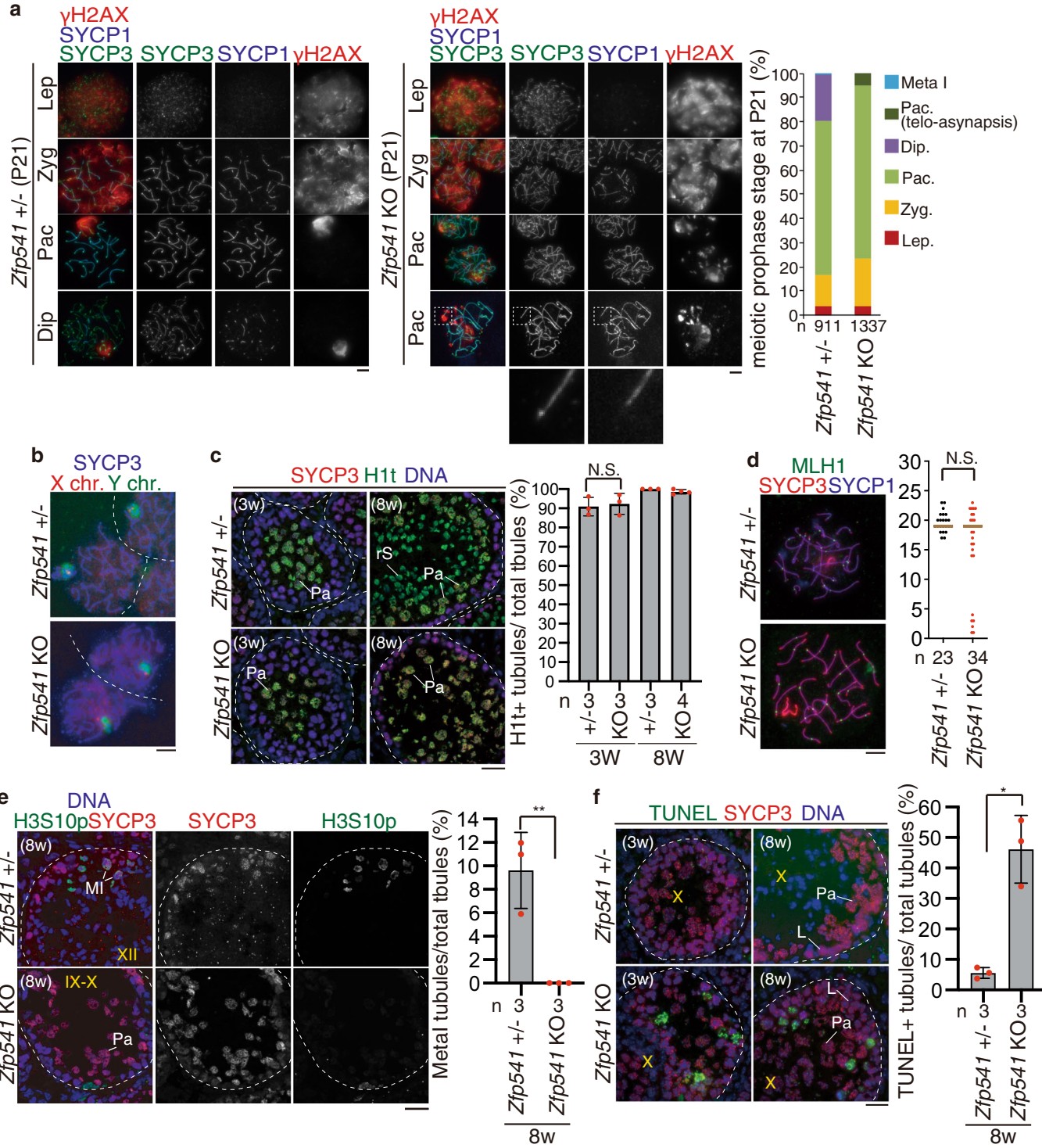

than pachytene did not appear in *Zfp541* KO mice at P21 (Fig. 3a), whereas the first wave of spermatogenesis passed through the meiotic prophase in the age-matched control. Notably, γH2AX still remained along synapsed autosomes despite the completion of homolog synapsis, and 6.9% of pachytene nuclei exhibited homolog chromosomes partially lacking SYCP1 assembly at the telomeric region in *Zfp541* KO at P21 (Fig. 3a). This suggested that DNA damage response still remained active along synapsed homologs in *Zfp541* KO. In addition to the aberrant morphology of the pachytene nuclei, γH2AX signal was diffusively observed on the sex chromosomes in some *Zfp541* KO spermatocytes (Fig. 3a).

Immuno-FISH indicated that X and Y chromosomes were apparently paired in most of the *Zfp541* KO pachytene spermatocytes (Fig. 3b). These observations suggested that heterochromatic XY body formation was defective in *Zfp541* KO spermatocytes, albeit apparently normal X and Y chromosome pairing (Fig. 3b). Close inspection of the seminiferous tubules indicated that *Zfp541* KO spermatocytes reached at least late pachytene as indicated by the presence of H1t staining (Fig. 3c). The pachytene spermatocytes in *Zfp541* KO were comparable to the control up to stage IX (Supplementary Fig. 4). The number of MLH1 foci (a marker of crossover recombination) in *Zfp541* KO pachytene spermatocytes

**Fig. 3 Zfp541 knockout spermatocytes fail to complete meiotic prophase. a** Chromosome spreads of *Zfp541*± and *Zfp541* KO spermatocytes at postnatal day 21 (P21) were immunostained as indicated. Enlarged images are shown to highlight asynaped chromosomes in pachytene-like cell. Lep: leptotene, Zyg: zygotene, Pac: pachytene, Dip: diplotene. Quantification of meiotic prophase stage spermatocytes per total SYCP3+ spermatocytes in *Zfp541*± and *Zfp541* KO mice is shown on the right. *n*: the number of cells examined. Scale bars: 25 μm. **b** XY chromosome pairing in *Zfp541*± (*n* = 52) and *Zfp541* KO (*n* = 50) pachytene spermatocytes was examined by immune-FISH using Y painting probe (green) and X point probe (red). X and Y chromosomes were in close proximity in most of the *Zfp541* KO pachytene spermatocytes that we examined (100% = 52/52 in the control, 98% = 49/50 in *Zfp541* KO, Chi-square test: *p* = 0.1474). Scale bar: 5 μm. **c** Seminiferous tubule sections (3-weeks and 8-weeks old) were stained for SYCP3, H1t, and DAPI. Pa: pachytene spermatocyte, rS: round spermatid. Shown on the right is the quantification of the seminiferous tubules that have H1t+/SYCP3+ cells per the seminiferous tubules that have SYCP3+ spermatocyte cells in *Zfp541*± and *Zfp541* KO mice (mean with SD). *n*: the number of animals examined for each genotype. Statistical significance is shown (two-tailed *t*-test). N.S.: Statistically not significant. Scale bar: 25 μm. **d** Chromosome spreads of *Zfp541*± and *Zfp541* KO spermatocytes were stained for MLH1, SYCP3, and SYCP1. The number of MLH1 foci is shown in the scatter plot with median (right). Statistical significance is shown (two-tailed Mann–Whitney *U*-test). *n*: number of spermatocytes examined. N.S.: Statistically not significant. Scale bar: 5 μm. **e** Seminiferous tubule sections (8-weeks old) were stained for SYCP3, H3S10p, and DAPI. M I: Metaphase I spermatocyte. Shown on the right is the quantification of the seminiferous tubules that have H3S10p+/SYCP3+ Metaphase I cells per the seminiferous tubules that have SYCP3+ spermatocyte cells in *Zfp541*± and *Zfp541* KO testes (mean with SD). *n*: the number of animals examined for each genotype. Statistical significance is shown by *p = 0.0068 (two-tailed *t*-test). Scale bar: 25 μm. **f** Seminiferous tubule sections from 3-weeks and 8-weeks old mice were subjected to TUNEL assay with immunostaining for SYCP3. L: leptotene, Pa: pachytene. Shown on the right is the quantification of the seminiferous tubules that have TUNEL+ cells per total tubules in *Zfp541*± (8w; *n* = 3) and *Zfp541* KO (8w; *n* = 3) testes (mean with SD). Statistical significance is shown by *p = 0.0165 (two-tailed *t*-test). Scale bar: 25 μm.

was comparable to age-matched control (Fig. 3d), suggesting that crossover recombination was complete in some if not all *Zfp541* KO pachytene spermatocytes, even though DNA damage response was still active. It should be mentioned that the MLH1 foci showed bimodal distribution in *Zfp541* KO. Although we do not know the exact reason, the meiotic recombination process might have been compromised before the MLH1 foci appear in the subfraction of *Zfp541* KO pachytene spermatocytes. These observations suggested that the progression of the meiotic prophase beyond pachytene was compromised in *Zfp541* KO. Histone H3 Ser10 phosphorylation (H3S10p) marks the centromeric region at diplotene and the whole chromosome at metaphase I. In the control spermatocytes, spermatocytes at metaphase I were indicated by co-immunostaining of H3S10p and centromeric SYCP3 in the stage XII seminiferous tubules (Fig. 3e). In contrast, the stages XI–XII tubules of *Zfp541* KO that contain H3S10p + diplotene or Meta I spermatocytes were not observed (Fig. 3e), suggesting that *Zfp541* KO spermatocytes failed to reach diplotene. This idea was further supported by co-immunostaining of MEIKIN (late pachytene–Meta I kinetochore marker)[32] and centromeric SYCP3. In the control spermatocytes, spermatocytes at metaphase I were identified by MEIKIN+/centromeric SYCP3 signals (Supplementary Fig. 4c). However, tubules that contain spermatocytes with such MEIKIN+/centromeric SYCP3 were absent in *Zfp541* KO, indicating *Zfp541* KO spermatocytes failed to reach metaphase I. Hence, the primary defect had already occurred before the completion of the meiotic prophase rather than during spermatid development. Notably, in *Zfp541* KO, TUNEL-positive cells were observed in the stage X seminiferous tubules that contained late pachytene spermatocytes (Fig. 3f). This phenotype persisted until adulthood in *Zfp541* KO testes, as shown by a higher number of TUNEL-positive seminiferous tubules (~46% in total tubules) in *Zfp541* KO testes at 8-weeks old (Fig. 3f). These observations suggested that *Zfp541* KO spermatocytes failed to complete meiotic prophase and were consequently eliminated by apoptosis before diplotene. These results indicate ZFP541 is required for the completion of the meiotic prophase and the transition to the meiotic division phase and spermiogenesis.

**ZFP541 forms a complex with KCTD19 and HDAC1/2.** In order to elucidate the function of ZFP541, the interacting factors of ZFP541 were screened by immunoprecipitation (IP) followed by mass spectrometry (MS) analysis. Our ZFP541 IP-MS analysis demonstrated that ZFP541 associated with histone deacetylases HDAC1, HDAC2, POZ/BTB domain-containing KCTD19,

and terminal deoxynucleotidyltransferase-interacting factor 1 (TDIF1) in the chromatin-bound fractions of testes extracts (Fig. 4a and Supplementary Fig. 5), which was consistent with a previous study[28]. Similar results were reproducibly obtained by MS analysis of ZFP541-IP from the chromatin-unbound fraction and the MNase-released nucleosome fractions of testes extracts (Supplementary Fig. 5). Furthermore, reciprocal IP by KCTD19 antibody indicated that ZFP541, HDAC1, HDAC2, and TDIF1 were co-immunoprecipitated with KCTD19 (Fig. 4a and Supplementary Fig. 6). These results suggest that those factors form a complex and may play a role in transcriptional repression through histone deacetylases. It was previously shown that TDIF1[33–35] associated with histone deacetylases and ELM2-SANT domain-containing MIDEAS/ELMSAN1, and knockdown of TDIF1 led to mitotic chromosome misalignment[36]. However, the exact function of KCTD19 has remained elusive. RT-PCR analyses demonstrated that the *Kctd19* gene showed a specific expression in adult testes, and not in other adult organs we examined (Fig. 4b and Supplementary Fig. 8a), suggesting that *Kctd19* is a germ-cell-specific factor. Spermatogenic expression of the *Kctd19* gene was further confirmed by the reanalysis of previous scRNA-seq data of adult mouse testis[29]. The result indicated that *Kctd19* was coordinately expressed with *Zfp541* and the landmark genes of meiosis such as *Spo11*, along the pseudotime of spermatogenic development (Supplementary Fig. 1). Strikingly, immunostaining of the spread chromosome indicated that KCTD19 protein was expressed in the nuclei of the spermatocytes from pachytene onward, and in round spermatids, but not in elongated spermatids (Fig. 4c). KCTD19 was co-expressed with its binding partner ZFP541 in testis (Fig. 4d). Close inspection by immunostaining of the seminiferous tubules indicated that the KCTD19 signal started to appear in the pachytene spermatocyte nuclei at the earliest in seminiferous stage II-III (Fig. 4e and Supplementary Fig. 7). KCTD19 signal persisted in round spermatids of stage VII seminiferous tubules. However, the KCTD19 signal was not observed in spermatocytes before pachytene, elongated spermatids, or spermatogonia. It should be mentioned that the KCTD19 signal was still not observed in H1t-negative early pachytene cells of stage I seminiferous tubules (Fig. 4f), suggesting that KCTD19 appeared in spermatocyte nuclei slightly later than ZFP541. In females, although *Kctd19* mRNA was detected in embryonic ovaries by RT-PCR (Supplementary Fig. 8a), protein expression of KCTD19 in embryonic ovaries was under detection limit by immunostaining (Supplementary Fig. 8b).

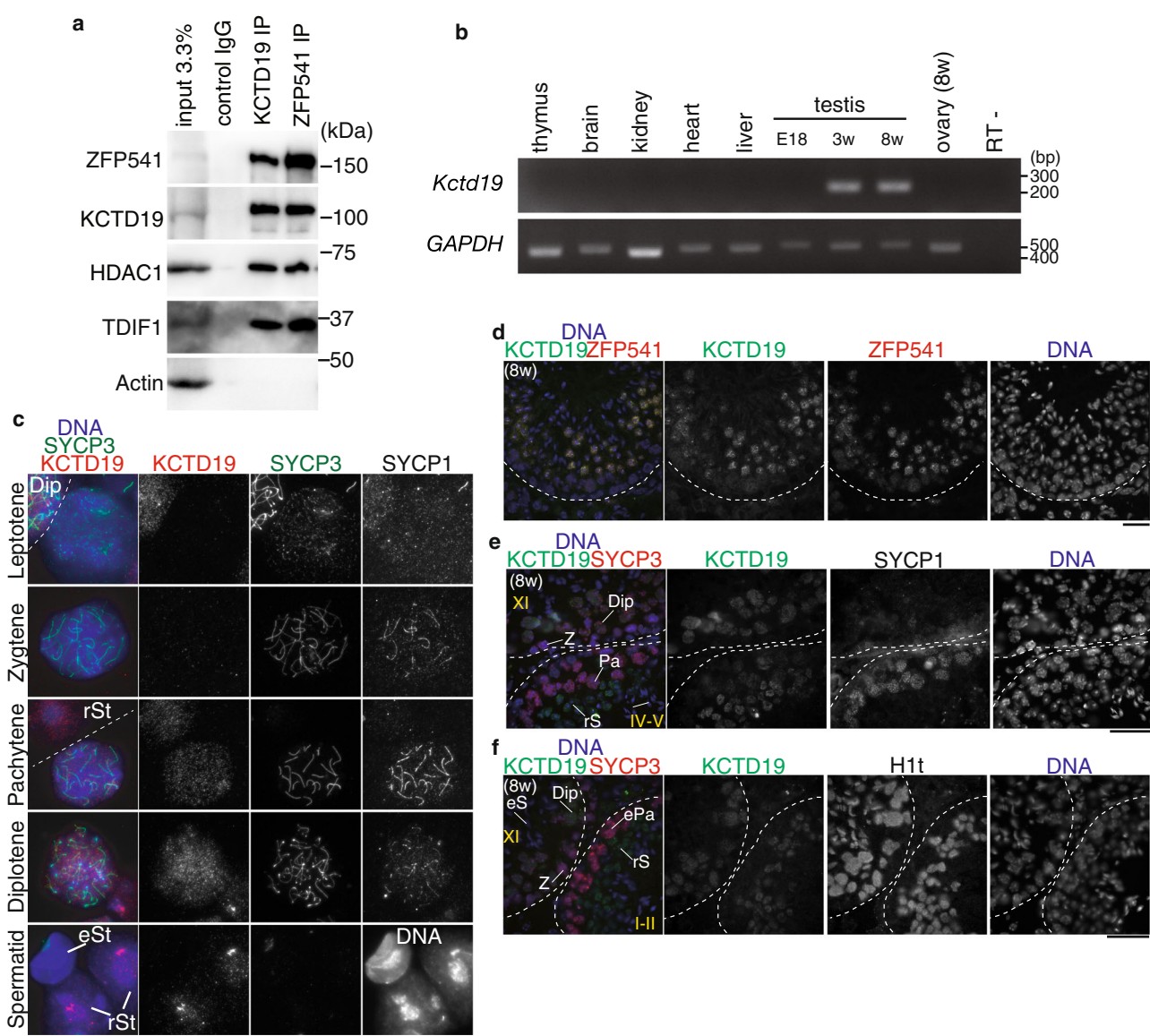

**Fig. 4 KCTD19 was identified as a ZFP541-interacting factor in the testis. a** Immunoblot showing the immunoprecipitates of ZFP541 and KCTD19 from chromatin extracts of WT mouse testes. The data were acquired from two separate experiments. **b** The tissue-specific expression pattern of *Kctd19* was examined using RT-PCR. Testis RNA was obtained from embryonic day 18 (E18), 3-weeks old (3w), and 8-weeks old (8w) male mice. Ovary RNA was obtained from adult 8-weeks old (8w) female mice. RT− indicates control PCR without reverse transcription. The data were acquired from two separate experiments. **c** Chromosome spreads of WT spermatocytes and spermatids were immunostained as indicated. Dip: Diplotene, rSt: round spermatid, eSt: elongated spermatid. Independent germ cells (at least *N* = 3) for each stage were examined in a single experiment. Scale bar: 5 μm. **d**–**f** Seminiferous tubule sections in WT testis (8-weeks old) were immunostained as indicated. Z: zygotene, Pa: pachytene, ePa: early pachytene, Dip: diplotene, rS: round spermatid, eS: elongated spermatid. Boundaries of the seminiferous tubules are indicated by white dashed lines. Roman numbers indicate the seminiferous tubule stages. Biologically independent mice (*N* = 2) were examined in two separate experiments. Scale bars: 25 μm for (**d**), 15 μm for (**e**), and (**f**).

***Kctd19* knockout in mice results in male infertility**. In order to address the role of KCTD19, we deleted Exon3–Intron12 of the *Kctd19* loci in C57BL/6 fertilized eggs by CRISPR/Cas9-mediated genome editing (Fig. 5a). In *Kctd19* KO testis, ZFP541 protein was expressed, and in *Zfp541* KO testes KCTD19 was expressed (Fig. 5b), implying that the expressions of ZFP541 and KCTD19 are independent of each other. Notably, whereas ZFP541 localized in the nuclei of the *Kctd19* KO seminiferous tubules, KCTD19 was hardly detected in the nuclei of the *Zfp541* KO seminiferous tubules despite the expression of KCTD19 in *Zfp541* KO testes, suggesting that nuclear localization of KCTD19 was impaired or KCTD19 protein was dispersed in the absence of ZFP541

(Fig. 5c). Similar to *Zfp541* KO mice, whereas the *Kctd19* KO females exhibited seemingly normal fertility with no apparent defect in adult ovaries (Supplementary Fig. 8c, d), *Kctd19* KO males exhibited smaller-than-normal testes (Fig. 5d). Indeed, histological analyses of *Kctd19* KO males revealed impaired spermatogenesis shown by post-meiotic spermatids and spermatozoa being absent in the seminiferous tubules and the caudal epididymis (Fig. 5e, f). Consistently, seminiferous tubules that contain PNA lectin-positive spermatids were absent in *Kctd19* KO (Supplementary Fig. 4a). These observations suggested that KCTD19 is required for the normal progression of spermatogenesis.

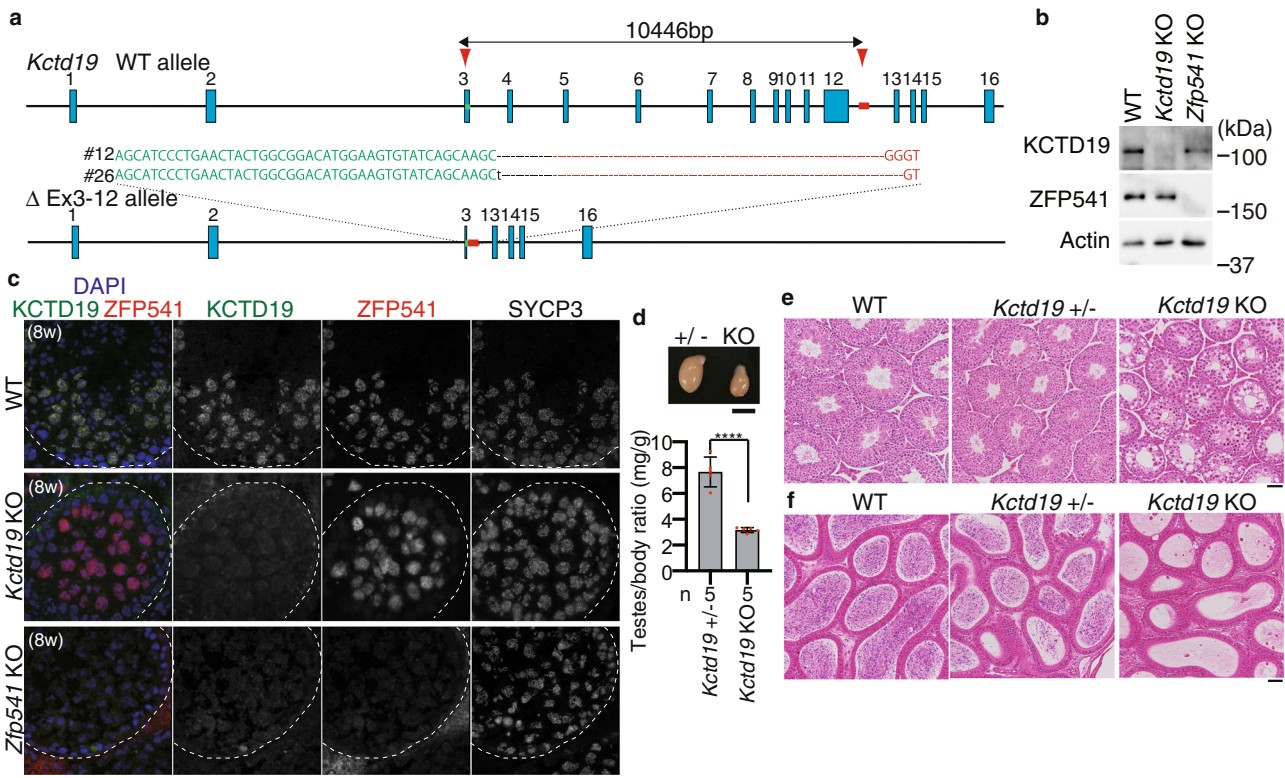

**Fig. 5 Spermatogenesis was impaired in *Kctd19* knockout male. a** The allele with targeted deletion of Exon3-12 in the *Kctd19* gene was generated by the introduction of CAS9, the synthetic gRNAs designed to target Exon 3, and the downstream of Exon12 (red arrowheads), and ssODN into C57BL/6 fertilized eggs. Two lines of KO mice were established. Line #12 of *Kctd19* KO mice was used in most of the experiments, unless otherwise stated. **b** Immunoblot analysis of whole testis extracts prepared from mice with the indicated genotypes (P21). Biologically independent mice (*N* = 2) were examined in two separate experiments. **c** Seminiferous tubule sections from WT, *Kctd19* KO, and *Zfp541* KO (8-weeks old) were immunostained as indicated. Biologically independent mice (*N* = 2) were examined in two separate experiments. Scale bar: 25 μm. **d** Testes from *Kctd19*± and *Kctd19* KO (8-weeks old). Scale bar: 5 mm. Testis/body-weight ratio (mg/g) of *Kctd19*± and *Kctd19* KO mice (8-weeks old) is shown below (mean with SD). *n*: number of animals examined. Statistical significance is shown by **$p$ = 0.0012 (two-tailed *t*-test). **e** Hematoxylin and eosin staining of the sections from WT, *Kctd19*±, and *Kctd19* KO testes (8-weeks old). Biologically independent mice (*N* = 3) for each genotype were examined. Scale bar: 50 μm. **f** Hematoxylin and eosin staining of the sections from WT, *Kctd19*±, and *Kctd19* KO epididymis (8-weeks old). Biologically independent mice (*N* = 3) for each genotype were examined. Scale bar: 50 μm.

**KCTD19 is required for meiotic prophase completion in male.**
*Kctd19* KO spermatocytes passed through late pachytene as indicated by the presence of H1t staining (Fig. 6a). However, we noticed that the number of MLH1 foci was reduced in *Kctd19* KO pachytene spermatocytes compared to the control (Fig. 6b), suggesting that crossover recombination was partly impaired in the absence of KCTD19. Furthermore, ~50% of *Kctd19* KO MLH1-positive pachytene spermatocytes were accompanied by asynapsed chromosomes (Fig. 6b), suggesting that homolog synapsis was partly defective in *Kctd19* KO. It should be mentioned that MLH1 foci showed bimodal distribution in *Kctd19* KO (Fig. 6b). Although we do not know the exact reason, the meiotic recombination process might have been compromised before the MLH1 foci appear in some, if not all, *Kctd19* KO. Consistently, the diplotene spermatocyte population was reduced compared to the control at P21, the time when the first wave of spermatogenesis reached diplotene and metaphase I in the control testis (Fig. 6c), suggesting that the progression of meiotic prophase was compromised in *Kctd19* KO. Notably, although *Kctd19* KO spermatocytes reached to metaphase I in adulthood in the stage XII seminiferous tubules (Fig. 6f), some, if not all, of metaphase I spermatocytes showed chromosome misalignment on the metaphase plate albeit with a normal number of bivalent chromosomes with chiasmata (Fig. 6d, e). Therefore, the meiotic

prophase defects in *Kctd19* KO derived at least in part from chromosome misalignment at metaphase I in the adult male. Furthermore, a subpopulation (31.9%) of the metaphase I cells was TUNEL positive in the stage XII seminiferous tubules of the *Kctd19* KO (Fig. 6g), suggesting that metaphase I cells were eliminated at least in part by apoptosis. Altogether, these results suggested that KCTD19 was required for the completion of meiosis I.

It is worth noting that *Kctd19* KO and *Zfp541* KO mice did not necessarily exhibit the same phenotypes, even though ZFP541 and KCTD19 formed a complex. Whereas spermatocytes beyond pachytene were absent in *Zfp541* KO (Fig. 3a), those that reached metaphase I appeared in *Kctd19* KO testis (Fig. 6d, e). Thus, *Zfp541* KO spermatocytes showed severer phenotype than *Kctd19* KO in terms of meiotic prophase progression. Given that ZFP541 remained in the nuclei in *Kctd19* KO whereas both ZFP541 and KCTD19 were absent in the nuclei of *Zfp541* KO spermatocytes, ZFP541 may have KCTD19-dependent and independent functions.

**Transcriptome analyses of *Zfp541*KO spermatocytes.** Transcriptome analysis on the spermatocytes was performed to address whether the observed phenotype in *Zfp541* KO testes was accompanied by alteration in gene expression. Although the cellular composition of testes changes during development, we assumed the

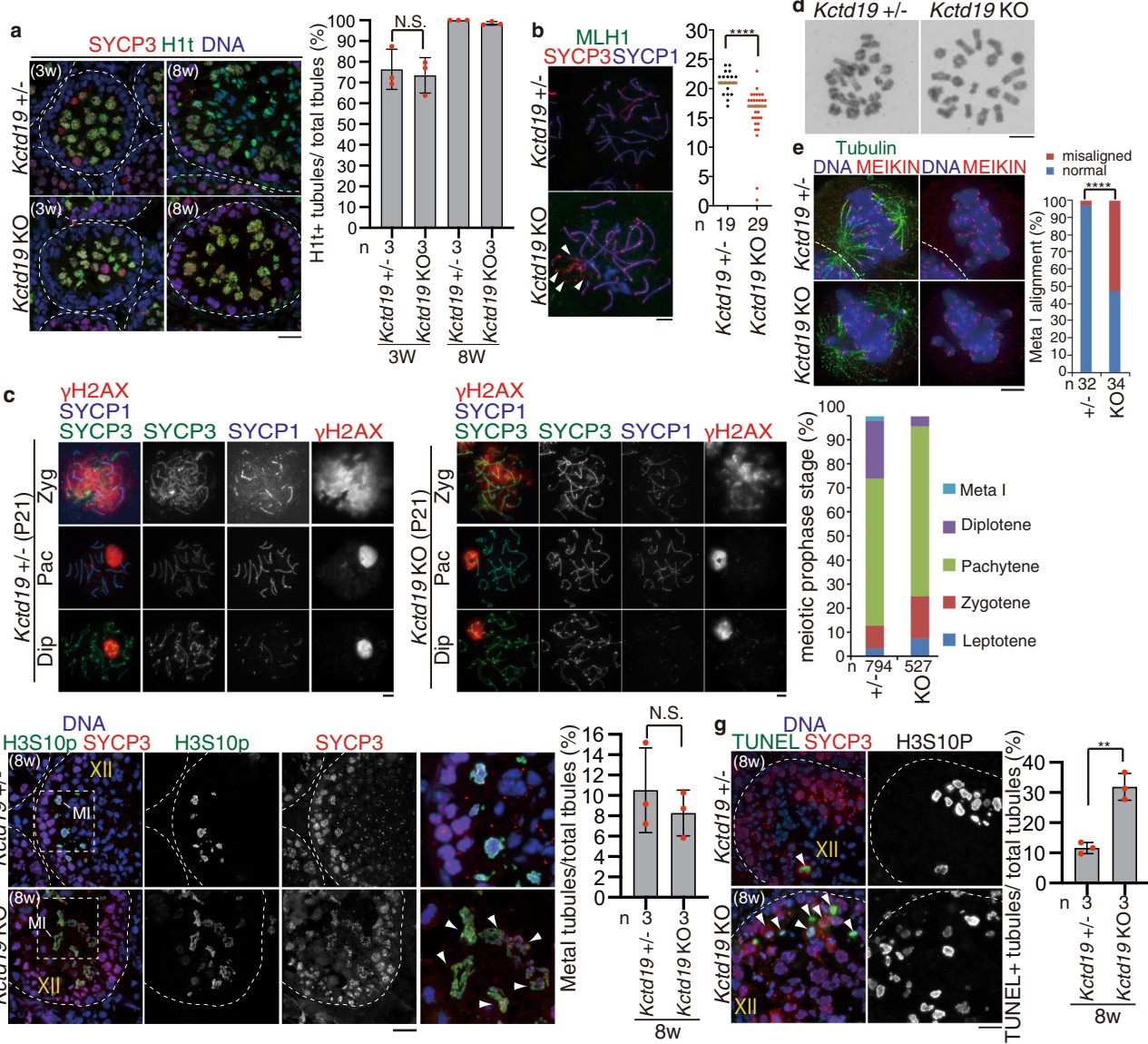

**Fig. 6 Kctd19 knockout spermatocytes fail to complete meiosis I. a** Seminiferous tubule sections (3-weeks and 8-weeks old) were immunostained as indicated. Shown on the right is the quantification of the seminiferous tubules that have H1t+/SYCP3+ cells per the seminiferous tubules that have SYCP3+ spermatocyte cells in *Kctd19±* and *Kctd19* KO mice (mean with SD). *n*: the number of animals examined for each genotype. Statistical significance is shown (two-tailed *t*-test). N.S.: statistically not significant. Scale bar: 25 μm. **b** Chromosome spreads of *Kctd19±* and *Kctd19* KO spermatocytes were immunostained as indicated. The number of MLH1 foci is shown in the scatter plot with median (right). Statistical significance is shown by *p*-value (two-tailed Mann–Whitney *U*-test). ****p < 0.0001. *n*: number of spermatocytes examined. Asynapsed chromosomes in MLH1+ pachytene spermatocytes are indicated by white arrowheads. Scale bar: 5 μm. **c** Chromosome spreads of *Kctd19±* and *Kctd19* KO spermatocytes at P21 were immunostained as indicated. Zyg: zygotene, Pac: pachytene, Dip: diplotene. Quantification of meiotic prophase stage per total SYCP3+ spermatocytes of *Kctd19±* and *Kctd19* KO mice is shown on the right. *n*: the number of cells examined. Scale bar: 5 μm. **d** Giemza staining of metaphase I chromosomes from *Kctd19±* and *Kctd19* KO spermatocytes. Independent Metaphase I cells (at least *N* = 19) for each genotype were examined in a single experiment. Scale bar: 5 μm. **e** Immunostaining of squashed metaphase I spermatocytes from *Kctd19±* and *Kctd19* KO. The metaphase I kinetochores were identified by MEIKIN immunostaining. Quantification of metaphase I spermatocytes with chromosome misalignment is shown on the right. *n*: the number of spermatocytes examined. Statistical significance is shown by ****p = 2.65 × 10^{-10} (chi-square test). Scale bar: 5 μm. **f** Seminiferous tubule sections (8-weeks old) were stained for SYCP3, H3S10p, and DAPI. M I: Metaphase I spermatocyte. Enlarged images are shown in the rightmost panels. Arrowheads indicate misaligned chromosomes. Shown on the right is the quantification of the seminiferous tubules that have H3S10p+/SYCP3+ Metaphase I cells per the seminiferous tubules that have SYCP3+ spermatocyte cells in *Kctd19±* and *Kctd19* KO testes (mean with SD). *n*: the number of animals examined for each genotype. Statistical significance is shown by *p*-value (two-tailed *t*-test). N.S.: Statistically not significant. Scale bar: 25 μm. **g** Seminiferous tubule sections from 8-weeks old mice were subjected to TUNEL assay with immunostaining for SYCP3 and H3S10p. Arrowheads indicate TUNEL-positive cells. Shown on the right is the quantification of the seminiferous tubules that have TUNEL+ cells per total tubules in *Kctd19±* and *Kctd19* KO testes (mean with SD). Statistical significance is shown by **p = 0.0019 (two-tailed *t*-test). Scale bar: 25 μm. Boundaries of the seminiferous tubules are indicated by white dashed lines.

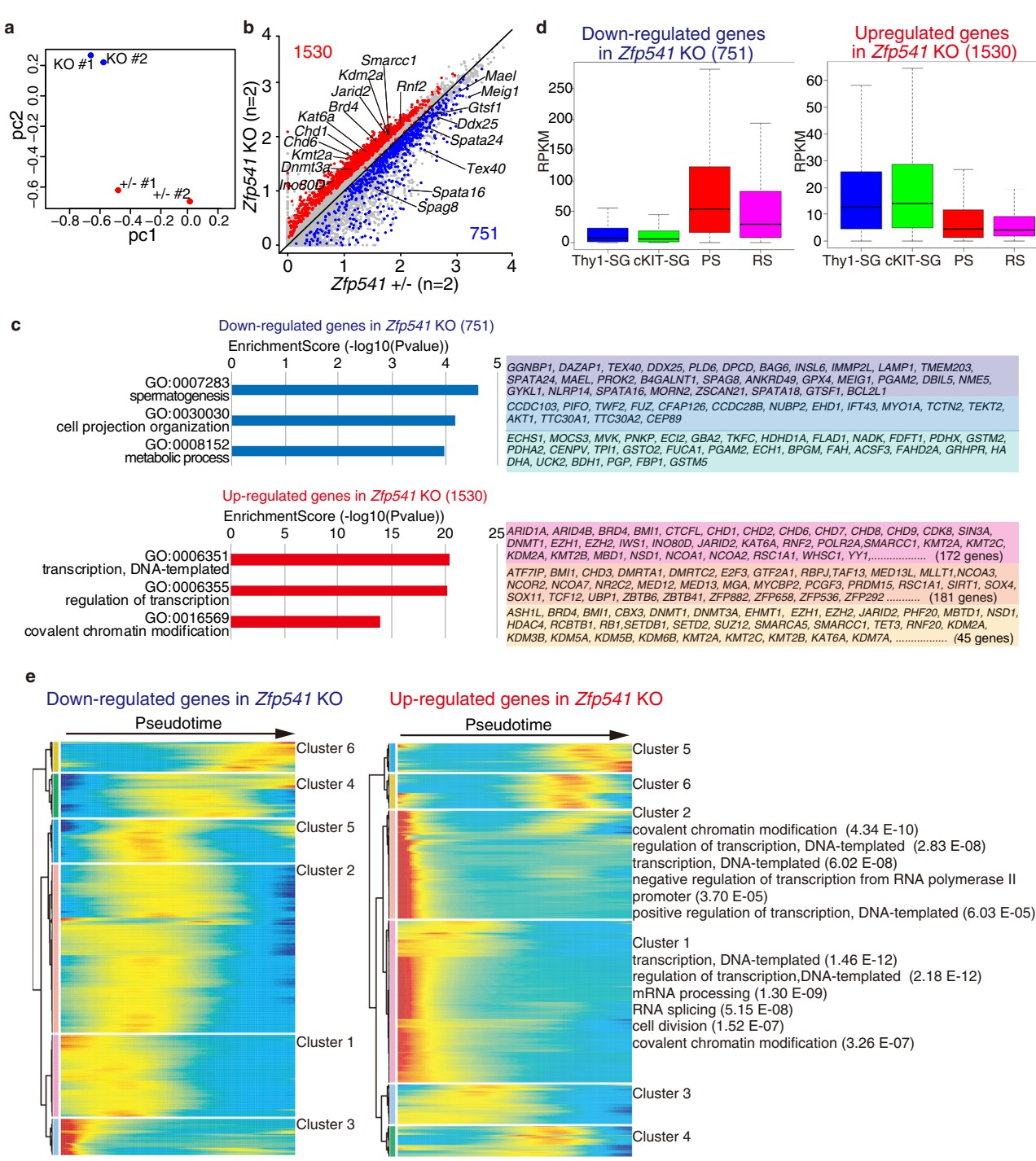

first wave of spermatocyte should progress with the same cellular composition in the control *Zfp541+/−* and *Zfp541* KO juvenile testes until the defects appear in the mutants. Since ZFP541 appeared from pachytene onward and *Zfp541* KO showed defects at the later stage of pachytene but before diplotene, we analyzed the transcriptomes of H1t-positive pachytene enriched population that represents the stage when the defects started to emerge in the mutant testes. For this purpose, we generated *Rec8-3xFLAG-HA-p2A-GFP* knock-in (*Rec8-3FH-GFP* KI) mice (Supplementary Fig. 9), as REC8 is expressed exclusively during meiotic prophase, which culminates at pachytene[37]. The meiotic prophase-enriched population (~50% of GFP-positive cells consist of H1t-positive

pachytene spermatocytes) was isolated from the control *Zfp541+/−* and *Zfp541* KO testes at P18 in *Rec8-3FH-GFP* KI background by fluorescent sorting of GFP-positive cells (Supplementary Fig. 9e, f). As mentioned earlier, since this is the stage before the defects appear in the mutants, we assumed the cellular composition should be similar in the control and *Zfp541* KO at this stage. This allowed the comparison of the transcriptomes of GFP-expressing meiotic prophase cells in the control *Zfp541±* and *Zfp541* KO by RNA-seq (Fig. 7) with a minimized batch effect that could potentially be made by a bias in the cellular population.

Principal component analysis (PCA) revealed that the overall transcriptomes of the GFP-positive cells in *Zfp541* KO testes were

**Fig. 7 Transcriptome analysis of meiotic prophase spermatocytes in *Zfp541* KO. a** The GFP-positive cells were isolated from control *Zfp541±* and *Zfp541* KO testes (P18) on the *Rec8-3FH-p2A-GFP* KI background by fluorescent sorting. Principal component analysis of the transcriptomes of GFP-expressing cells (meiotic prophase spermatocytes) in *Zfp541±* and *Zfp541* KO on the *Stra8-3FH-p2A-GFP* KI background is shown. **b** Scatter plot of the transcriptome of GFP-expressing cells in *Zfp541±* ($n = 2$) versus *Zfp541* KO ($n = 2$) is shown. The numbers of differentially expressed genes are shown. Significance criteria: false discovery rate ≤ 0.05. **c** GO analysis of the downregulated genes (upper) and upregulated genes (lower) in GFP-expressing cells of *Zfp541* KO testes. See Supplementary Data 1 for a complete gene list of the GO analyses. **d** Expression levels (RPKM) of the downregulated 751 genes (left) and the upregulated 1530 genes (right) in GFP-expressing spermatocytes of *Zfp541* KO are shown by box-whisker plot (whiskers indicate min and max. Bounds of box indicate 25th and 75th percentiles quantile with median). The upregulated and downregulated genes in *Zfp541* KO were reanalyzed with the previous y published data of stage-specific bulk RNA-seq ($N = 2$ biologically independent samples)[22,24]. Thy-SG: Thy+ spermatogonia, cKIT-SG: cKIT+ spermatogonia, PS: pachytene spermatocytes, RS: round spermatid. **e** Heatmaps showing the hierarchical relationship among the clusters of DEGs across pseudotime of spermatogenesis. Expressions of the downregulated genes (left) and the upregulated genes (right) in GFP-expressing spermatocytes of *Zfp541* KO was assessed by reanalyzing scRNA-seq data of spermatogenic cells (GEO: GSE109033)[29]. Pseudotime (left to right) corresponds to the developmental trajectory of spermatogenesis (undifferentiated spermatogonia to round spermatids). In clusters 1 and 2 of upregulated genes, GO terms (top 5 by *p*-value) with false discovery rate (FDR) < 0.01 are shown. In other clusters, GO-term is not shown due to statistically higher FDR. See Supplementary Data 2 for the complete gene list of the GO analyses.

different from those of the control *Zfp541±* testes at P18 (Fig. 7a, b). Notably, gene ontology (GO) analysis of differentially expressed genes (DEG) between P18 control *Zfp541±* and *Zfp541* KO testes indicated that the genes involved in transcriptional regulation and covalent chromatin modification were upregulated in *Zfp541* KO (Fig. 7c and Supplementary Data 1). Reanalysis of the data together with the data from previous studies on transcriptomes during spermatogenesis[22,24] demonstrated that the upregulated genes in *Zfp541* KO (1530 genes) were overall expressed in spermatogonia and the expression declined from prophase onward (Fig. 7d). In contrast, the downregulated genes in *Zfp541* KO (751 genes), were overall less expressed in spermatogonia and highly expressed in prophase spermatocytes (Fig. 7d). This idea was further supported by reanalyzing the DEGs using previously published scRNA-seq data of spermatogenic cells[29] (Fig. 7e and Supplementary Data 2). The upregulated genes in *Zfp541* KO were overall found in early pseudotime. Among upregulated DEGs, 67.0% of the genes were in clusters 1 and 2, representing the highest expression in spermatogonia. In contrast, the downregulated genes in *Zfp541* KO were found around the mid to later stages of pseudotime. Notably, GO analysis demonstrated that the genes involved in spermatogenesis were downregulated in *Zfp541* KO (Fig. 7c and Supplementary Data 1), which at least in part accounts for the aforementioned cytological observation showing that *Zfp541* KO spermatocytes failed to complete the meiotic prophase. We cannot formally exclude the possibility that subtle cellular population difference between control *Zfp541±* and *Zfp541* KO testes may potentially contribute to bulk transcriptomes. Nevertheless, these results suggest that ZFP541 is required for the completion of the meiotic prophase program in male.

**ZFP541 binds to the promoter regions of the genes associated with transcriptional regulation.** ZFP541 is assumed to possess putative DNA-binding domains and associate with histone deacetylases, implying it plays a role in the regulation of transcription via the modulation of chromatin status. So we further investigated the ZFP541-target sites on the genome by chromatin immunoprecipitation followed by sequencing (ChIP-seq) analysis. ZFP541-binding sites were defined by commonly identified sites in two independent ChIP-seq data sets using two different antibodies (Supplementary Fig. 10a–c). As a result, ZFP541 bound to 6135 sites (5921 nearest genes, which were assigned regardless of the distance from the ZFP541-binding sites), of which 32.3% and 26.1% resided around the gene promoter regions and the 5′UTR on the mouse genome, respectively (Fig. 8a). Overall, ZFP541-binding sites resided within ±2 kb of the TSS (4689 genes) (Fig. 8b and Supplementary Fig. 10b, c, Supplementary Data 3). Crucially, ZFP541-binding sites well overlapped with the 5′-capped

sequence of mRNA, which were revealed by Cap analysis gene expression (CAGE)-seq in P10.5 testis[38]. Notably, DNA-binding motif analysis indicated that ZFP541-ChIP enriched GC-rich DNA sequences (Fig. 8c), suggesting that ZFP541 directly binds to the promoters through these motifs. Furthermore, the ZFP541-bound sites largely showed increased repressive histone mark H3K27me3 and reciprocally decreased active histone H3K27ac during the transition from prophase to round spermatids[14,24] (Supplementary Fig. 10d), suggesting that ZFP541-bound sites overall undergo repressive states after meiotic prophase. It should be mentioned that ZFP541-bound sites were less presented in the X and Y chromosomes, consistent with the immunostaining of ZFP541 that showed absence of staining in the XY body (Supplementary Fig. 10e, f).

Among the putative ZFP541-bound nearest genes, 841 and 213 genes were identified within the upregulated genes and the downregulated genes in *Zfp541* KO GFP-positive cells, respectively (Fig. 8d), supporting the idea that ZFP541 regulates the expressions of those target genes. Remarkably, GO analysis revealed that the putative ZFP541-target genes upregulated in *Zfp541* KO (841 genes) such as *Dnmt1*, *Dnmt3A* (DNA methyltransferase), *Ino80D*, *Chd1*, *Chd2*, *Chd3*, *Chd6* (chromatin remodeling), *Kat6A*, *Kmt2C*, *Kmt2B*, *Kdm2A*, *Kdm5B*, *Ash1L*, *Bmi1*, *Jarid2*, *Ezh1*, *Ezh2*, *Ehmt1*, *Rnf2*, *Suz12* (Histone modification), and *Ctcfl* (chromatin binding) were associated with the biological processes of transcriptional regulation and covalent chromatin modification (Fig. 8e and Supplementary Data 3). It should be noted that most of these genes were generally expressed in broad cell types rather than being germ-cell specific. In contrast, among the ZFP541-target genes downregulated by *Zfp541* KO (213 genes), we did not find GO terms that showed statistical significance (Supplementary Data 3).

Since ZFP541 was associated with histone deacetylases, we analyzed the ZFP541-bound genes along with the RNA expression and histone modification levels during spermatogenesis using the data set from previous studies[22,24]. Strikingly, overall expression levels of those putative ZFP541-target genes that were upregulated in *Zfp541* KO (841 genes) were declined in prophase spermatocytes and round spermatids (Fig. 8f). This was further supported by reanalyzing those genes with previously published scRNA-seq data of spermatogenic cells[29] as described above. 65.2% of the upregulated genes in *Zfp541* KO (clusters 1 and 4) were found in early pseudotime (Fig. 8g and Supplementary Data 4). Accordingly, overall H3K27me3 levels were increased across the gene body in prophase spermatocytes and remained high at the promoter regions in round spermatids (Fig. 8h and Supplementary Fig. 10d). This suggests that the HDAC1/2-containing ZFP541 complex represses the transcription of a

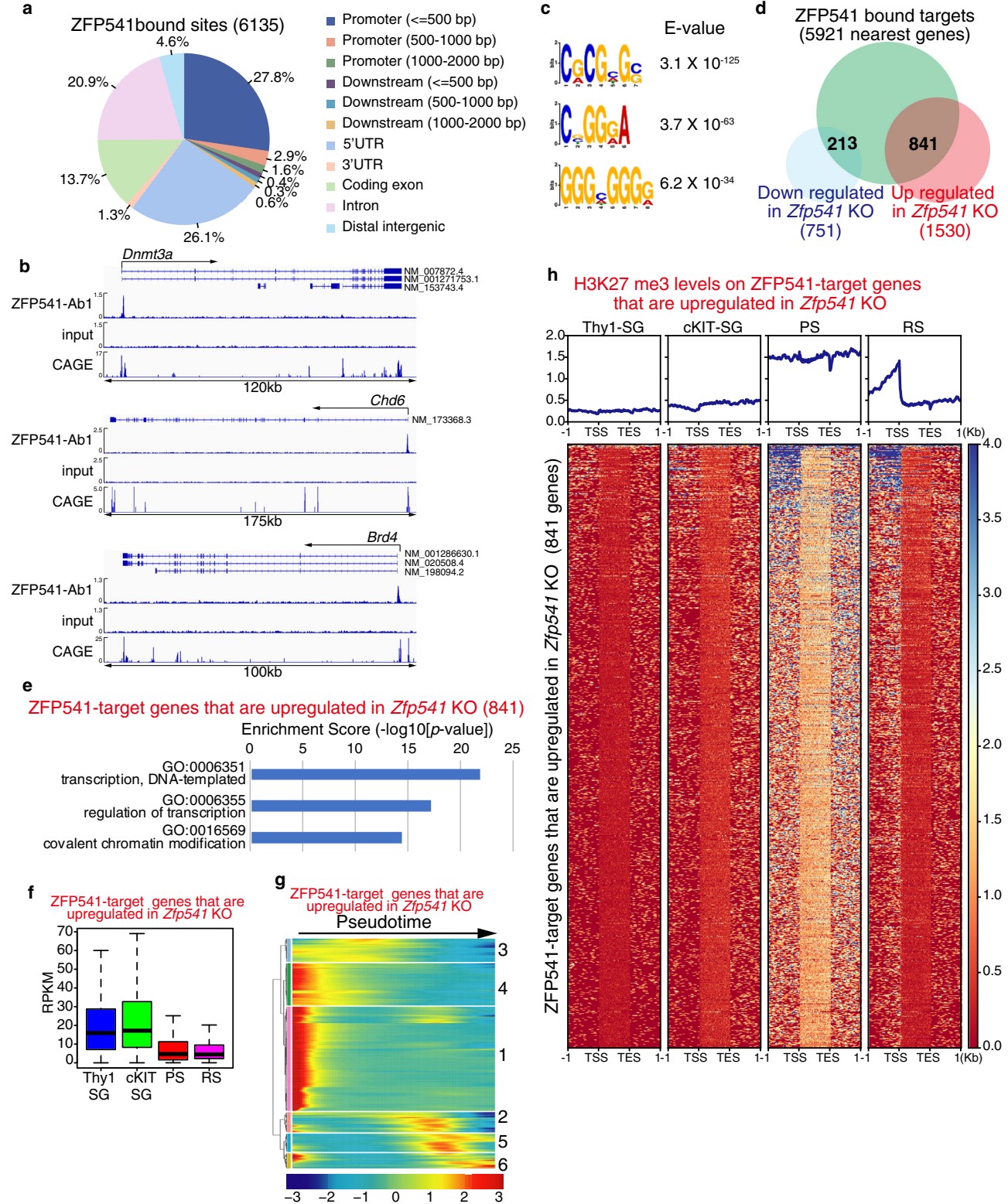

subset of critical genes that are involved in a broad range of transcriptional regulation and chromatin modification prior to the completion of meiotic prophase. Since a subset of those DEGs act for general transcription processes, we do not exclude the possibility that some, if not all, of the DEGs are indirectly regulated by ZFP541. It is also possible that DEGs putatively bound by ZFP541 are co-regulated by other transcription factors and/or epigenetic mechanisms. Nevertheless, these findings suggest that the ZFP541 complex at least in part binds to and regulates the expressions of a subset of genes for the progression of meiotic prophase towards completion (Fig. 9).

## Discussion

During spermatogenesis, meiosis is followed by spermiogenesis, which is accompanied by robust alterations of transcriptional

**Fig. 8 ZFP541 binds to the gene promoter regions. a** ZFP541-binding sites were classified by the genomic locations as indicated. **b** Genomic view of ZFP541 ChIP-seq and input DNA data over representative gene loci. Genomic coordinates were obtained from RefSeq. RefSeq IDs for mRNA isoforms are indicated. To specify testis-specific TSS, RNA-seq of the 5' capped end of the mRNA (CAGE) in P10.5 testis are shown[38]. **c** Top 3 sequence motifs enriched in ZFP541 ChIP-seq with E-values. MEME-ChIP E-value estimates the expected number of motifs with similar features that one would find in a similarly sized set of random sequences. **d** Venn diagram representing the overlap of ZFP541-bound genes (5921 nearest genes), downregulated (751 genes), and upregulated genes (1530 genes) in *Zfp541* KO mice. **e** GO analysis of the ZFP541-bound genes that were upregulated in *Zfp541* KO mice (841 genes) ($p <$ $1.0 \times 10^{-6}$). Top 3 biological processes ranked by log (*p*-value) are listed. The ZFP541-bound genes that were downregulated in *Zfp541* KO (213 genes) showed no GO terms statistically enriched. See Supplementary Data 3 for a complete gene list of the GO analyses. We used the standard parameters to detect over-represented GO terms for biological function, giving Fisher's exact *p*-values. **f** Expression levels (RPKM) of the ZFP541-bound genes that were upregulated in *Zfp541* KO spermatocytes (841 genes) are shown by box-whisker plot (whiskers indicate min and max. Bounds of box indicate 25th and 75th percentiles quantile with median). Thy-SG: Thy+ spermatogonia, cKIT-SG: cKIT+ spermatogonia, PS: pachytene spermatocytes, RS: round spermatid. **g** Heatmaps showing the hierarchical relationship among the clusters of the ZFP541-bound genes that were upregulated in *Zfp541* KO spermatocytes across pseudotime of spermatogenesis. Expressions of the genes were assessed by reanalyzing scRNA-seq data of spermatogenic cells as described in Fig. 5e[29]. Pseudotime (left to right) corresponds to the developmental trajectory of spermatogenesis (undifferentiated spermatogonia to round spermatids). The cluster number is indicated on the right. See Supplementary Data 4 for the complete gene list of clusters 1–6. **h** Heat map of H3K27me3 levels on the ZFP541-target genes that are upregulated in *Zfp541* KO (841 genes). H3K27me3 levels are shown on the genomic regions between −1.0 kb upstream of TSS and +1.0 kb downstream of TES. The color key is shown. Average distributions of H3K27me3 are shown (upper).

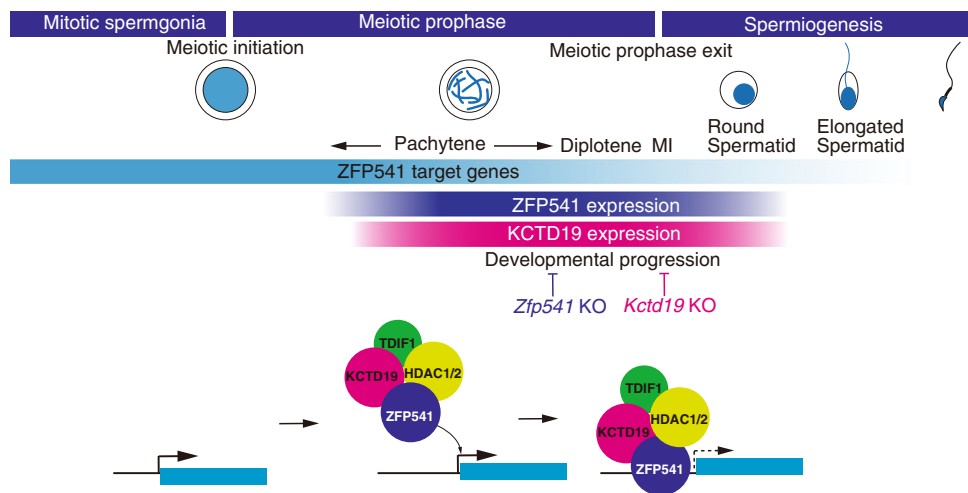

**Fig. 9 The ZFP541–KCTD19 complex is required for completion of the meiotic prophase.** Schematic model of the ZFP541–KCTD19–HDAC1/2-TDIF1-containing complex that suppresses a subset of genes for the completion of the meiotic prophase. The timing of ZFP541 and KCTD19 protein expressions are shown along the developmental stages. ZFP541 and KCTD19 started to appear in the spermatocyte nuclei from early pachytene onward, and were still expressed in round spermatids. Vertical bars indicate the stages when the developmental progression is blocked in *Zfp541*KO and *Kctd19* KO spermatocytes.

program and chromatin status to prepare for sperm production. In the present study, we demonstrate that ZFP541 forms a complex with KCTD19, HDAC and TDIF1 in spermatocytes and round spermatids (Fig. 4a and Supplementary Figs. 5, 6). Our genetic study demonstrated that both ZFP541 and KCTD19, whose expressions are germ-cell specific, are required for completion of meiotic prophase I in spermatocytes (Figs. 3 and 6), highlighting the sexual difference in the transcriptional program of meiotic prophase. It has been shown that androgen-dependent signaling from Sertoli cells peaks at stages VI–VII during the cycle of the seminiferous epithelium, and is required by spermatocytes for the exit from prophase and the entry into the division phase[25]. In the absence of the androgen receptor of the Sertoli cells (*Scra* KO), spermatocytes fail to acquire the competence to proceed into the meiotic division phase. Although we did not find any relationship between the peak expression of ZFP541 in spermatocytes and the androgen-dependent signaling from Sertoli cells, the response of the germ cells to extrinsic androgen signaling by Sertoli cells may coordinately function with the intrinsic gene expression of the germ cells to promote the completion of meiotic prophase and the subsequent first division phase. Interestingly, although ZFP541 and KCTD19 form a

complex, *Zfp541* KO and *Kctd19* KO mice exhibited different phenotypes (Figs. 3 and 6). Whereas spermatocyte beyond pachytene was absent in *Zfp541* KO (Fig. 3), metaphase I cells appeared in *Kctd19* KO testis (Fig. 6). Thus, *Zfp541* KO spermatocytes showed severer phenotype in meiotic prophase progression than *Kctd19* KO. Because both ZFP541 and KCTD19 were absent in the nuclei of *Zfp541* KO spermatocytes, whereas ZFP541 remained in the nuclei and was functional in *Kctd19* KO (Fig. 5c), ZFP541 may have a function independent of KCTD19.

It should be mentioned that MIDEAS and Transcription regulatory factor 1 (TRERF1), as well as TDIF1, were identified in KCTD19 immunoprecipitates (Supplementary Fig. 6b). Previously, it was shown that TDIF1, MIDEAS, and HDAC1 formed a mitotic deacetylase (MiDAC) complex in mitotic cells, and that loss of either TDIF1 or MIDEAS led to chromosome misalignment in mitosis[36]. Furthermore, MIDEAS and TRERF1 share homology in their ELM/SANT domains, and MIDEAS and TRERF1 interact with each other[36,39]. Given that MIDEAS and TRERF1, as well as ZFP541, possess ELM/SANT domains, KCTD19 may form a subcomplex that consists of MIDEAS, TRERF1, TDIF1, and HDAC1/2, which is distinct from the ZFP541–KCTD19–TDIF1–HDAC1/2

complex. Therefore, it is possible that the defect in metaphase I chromosome alignment in *Kctd19* KO was indirectly caused by the lack of MIDEAS–TRERF1–KCTD19–TDIF1–HDAC1/2 subcomplex, rather than ZFP541–KCTD19–TDIF1–HDAC1/2 complex (Fig. 6e).

Crucially, our ChIP-seq analysis indicates that ZFP541 binds to a broad range of genes associated with biological processes of transcriptional regulation and covalent chromatin modification, such as DNA methylation, chromatin remodeling, and histone modification (Fig. 8b, e) that are generally expressed in a broad range of cell types. Because ZFP541 associates with HDAC1 and HDAC2 (Fig. 4a and Supplementary Fig. 5), we reasoned that ZFP541 plays a role in repressing the target genes prior to the completion of the meiotic prophase program (Fig. 9). Consistent with this idea, the expression levels of the ZFP541-target genes were overall repressed and were accompanied by repressive H3K27me3 mark in round spermatids (Fig. 8f, g). Furthermore, it should be mentioned that most of the ZFP541-target genes, if not all, were well overlapped with the target genes of a germ-cell-specific PRC1 component SCML2 in germline stem cells (Supplementary Fig. 10g)[22–24]. Since SCML2 binds to somatic/progenitor genes in the stem cell stage to suppress these genes after meiosis[22], it is possible that those ZFP541-target genes might be pre-marked with SCML2 and Polycomb proteins for later suppression of transcription. Supporting this possibility, ZFP541-target genes are marked with PRC2-mediated H3K27me3 (Fig. 8h), which is established downstream of SCML2 in pachytene spermatocytes[24]. Therefore, ZFP541 may trigger the reconstruction of the transcription network to promote the completion of prophase, finalize meiotic divisions, and proceed into spermatid production, which may be similar to the function of the transcription factor Ndt80 that promotes pachytene exit and spore formation in budding yeast[40,41]. Our study sheds light on the regulatory mechanisms of gene expression that promote the developmental progression of meiotic prophase leading to spermatid differentiation.

## Methods

**Animal experiments**. *Zfp541* and *Kctd19* knockout mice were C57BL/6 background. *Rec8-3xFLAG-HA-p2A-GFP* knock-in mice were congenic with the C57BL/6 background. Whenever possible, each knockout animal was compared to littermates or age-matched non-littermates from the same colony, unless otherwise described. Animal experiments were approved by the Institutional Animal Ethics Committees of Kumamoto University (approval F28-078, A2020-006, A30-001, A28-026). *Zfp541* and *Kctd19* knockout mouse and *Rec8-3xFLAG-HA-p2A-GFP* knock-in mouse lines generated in this study have been deposited to Center for Animal Resources and Development (CARD) (ID 2857 for *Zfp541* respectively; ID 2879, ID 2880 for *Kctd19* Line #12, Line #26 respectively; ID2681 for *Rec8-3xFLAG-HA-p2A-GFP*).

**Generation of *Zfp541* knockout mice and genotyping**. *Zfp541* knockout mice were generated by introducing Cas9 protein (317-08441; NIPPON GENE, Toyama, Japan), tracrRNA (GE-002; FASMAC, Kanagawa, Japan), synthetic crRNA (FAS-MAC), and ssODN into C57BL/6N fertilized eggs using electroporation. For the generation of *Zfp541* Exon1–13 deletion (Ex1-13Δ) allele, the synthetic crRNAs were designed to direct CATGGAGCCATACAGTCTT(GGG) of the *Zfp541* exon 1 and ACGACCAAAGAAACAGTGCT(GGG) in the intron 3. ssODN: 5′-GAA CGAATTCACCTTCTAGCCAAAAACGACCAAAGAAACAGTatttatttACTGTA TGGCTCCATGGCTTTCCACTGCCAGGTCCTGGCT-3′ was used as a homologous recombination template.

The electroporation solutions contained [10 μM of tracrRNA, 10 μM of synthetic crRNA, 0.1 μg/μl of Cas9 protein, ssODN (1 μg/μl)] for *Zfp541* knockout in Opti-MEM I Reduced Serum Medium (31985062; ThermoFisher Scientific). Electroporation was carried out using the Super Electroporator NEPA 21 (NEPA GENE, Chiba, Japan) on Glass Microslides with round wire electrodes, 1.0 mm gap (45-0104; BTX, Holliston, MA). Four steps of square pulses were applied (1, three times of 3 mS poring pulses with 97 mS intervals at 30 V; 2, three times of 3 mS polarity-changed poring pulses with 97 mS intervals at 30 V; 3, five times of 50 mS transfer pulses with 50 mS intervals at 4 V with 40% decay of voltage per each

pulse; 4, five times of 50 mS polarity-changed transfer pulses with 50 mS intervals at 4 V with 40% decay of voltage per each pulse).

The targeted *Zfp541* Ex1-13Δ allele in F0 mice was identified by PCR using the following primers; Zfp541-F1: 5′-AGCTAGCTGCCAGCGAGGGCTCTTC-3′ and Zfp541-R3: 5′-GAGGCAGCAGAAGGGAGGTAGGATG-3′ for the knockout allele (404 bp). Zfp541-F1 and Zfp541-R2: 5′-GGTTGAGTGTGTCACTGCAGT TGAG-3′ for the Ex1 of WT allele (199 bp). Zfp541-F2: 5′-AACGTAGGAAGCA GATTTCAGGCGG-3′ and Zfp541-R1: 5′-TCCCTCAGCTGGGCCATCCAAGT CC-3′ for the Ex10-11 of WT allele (613 bp). The PCR amplicons were verified by Sanger sequencing. Primer sequences are listed in Supplementary Data 5.

**Generation of *Kctd19* knockout mice and genotyping**. *Kctd19* knockout mouse was generated as described above. For the generation of *Kctd19* Exon3-12 deletion (Ex3-12Δ) allele, the synthetic crRNAs were designed to direct GAAGTGTAT-CAGCAAGCCCT (CGG) of the *Kctd19* exon 3 and AAGAGGCCAATCATC-TAGGG (TGG) in the intron12. ssODN: 5′-ACCCTAGATGATTGGCCTCTTGGTGGCACCTCTGCCTCCC tttatttatt-caGCTTGCTGATACACTTCCATGTCCGCCAGTAGTTCAGGGATGCT-3′ was used as a homologous recombination template.

The targeted *Kctd19* Ex3-12Δ allele in F0 mice was identified by PCR using the following primers; Kctd19-F1: 5′-AGGCAGCACTCTTCTCTGTGGGACAG-3′ and Kctd19-R1: 5′-TTTCCCATGCCACCTGTGCCTTTCC-3′ for the knockout allele (279 bp). Kctd19-F1 and Kctd19-R2: 5′-TGCCCGTGACGGTACCTGTGA AGGC-3′ for the Ex3 of WT allele (192 bp). Kctd19-F2: 5′-GGCAAGCAGAGG CCCAAGGACAGAG-3′ and Kctd19-R1 for the intron12 of WT allele (714 bp). The PCR amplicons were verified by Sanger sequencing. Primer sequences are listed in Supplementary Data 5.

**Generation of *Rec8-3xFLAG-HA-p2A-GFP* knock-in mouse and genotyping**. The targeting vector was designed to insert 3xFLAG-HA-p2A-EGFP-3′UTR in-frame with the coding sequence into the Exon 20 of the *Rec8* genomic locus. Targeting arms of 1018 and 812 bp fragments, 5′ and 3′ of the Exon 20 of *Rec8* gene respectively, were generated by PCR from mouse C57BL/6 genomic DNA and directionally cloned flanking p*GK-Neo*-polyA and *DT-A* cassettes. The 5′ arm is followed by nucleotide sequences encoding 3xFLAG, HA, p2A, EGFP, and the 3′UTR of *Rec8* gene[42]. TT2 ES[42] cells were co-transfected with the targeting vector and pX330 plasmids (Addgene) expressing Crispr-gRNAs directing TATGTGTT CAGAGCTGAGAC(tgg) and GGAAGGGAGGGGCTGCGCTG(agg), which locates at the 3′ region of the Exon 20 of *Rec8* gene. The G418-resistant ES clones were screened for homologous recombination with the *Rec8* locus by PCR using primers Rec8_5arm_F1: 5′-GGAGCTCTTCAGAACCCCAACTCTC-3′ and KI96ES-19814R-HA: 5′-GGGCACGTCGTAGGGGTATCCCTTG-3′ for the left arm (1490 bp).

The homologous recombinant cells were isolated and chimeric mice were generated by aggregation (host ICR) of recombinant ES cells. Chimeric males were mated to C57BL/6N females and the progenies were genotyped by PCR using the primers Rec8-Left2-F: 5′-TGAGCAAGTTCATTAAACCATATCCTG-3′ and Rec8-Rightarm-R: 5′-CTCTTGAAGCTGACATCTTTGGTTAC-3′ for the knock-in allele (2844 bp) and the WT allele (1045 bp). Primer sequences are listed in Supplementary Data 5.

**PCR with reverse transcription**. Total RNA was isolated from tissues and embryonic gonads using TRIzol (ThermoFisher). cDNA was generated from total RNA using Superscript III (ThermoFisher) followed by PCR amplification using Ex-Taq polymerase (Takara) and template cDNA. For RT-qPCR, total RNA was isolated from WT (n = 3), *Meiosin* KO (n = 3), and *Stra8* KO (n = 3) testes, and cDNA was generated as described previously[26]. *Zfp541* cDNA was quantified by ΔCT method using TB Green Premix Ex Taq II (Tli RNaseH Plus) and Thermal cycler Dice (Takara), and normalized by *GAPDH* expression level.

Sequences of primers for RT-PCR are as follows:
Gapdh_F for Fig. 1c: 5′-TTCACCACCATGGAGAAGGC-3′
Gapdh_R for Fig. 1c: 5′-GGCATGGACTGTGGTCATGA-3′
Gapdh_F2(Gapdh5599) for Fig. 3b, Supplementary Fig. 8a: 5′-ACCACAGTCCATGCCATCAC-3′
Gapdh_R2(Gapdh5600) for Fig. 3b, Supplementary Fig. 8a: 5′-TCCACCACCCTGTTGCTGTA-3′
Zfp541_rtF2: 5′-CTGTTACTCAGAGGACCCCAGAAG-3′
Zfp541_rtR1: 5′-ATCTTTAACTCAGGAGTTGGACGG-3′
KCTD19_rtL1: 5′-GAAGACAATCTGCTGTGGCTG-3′
KCTD19_rtR1: 5′-GGGAACAGCCACCCATATTCA-3′
Primer sequences are listed in Supplementary Data 5.

**Generation of antibodies**. His-tagged recombinant proteins of ZFP541-N (aa 1–517), KCTD19-N (aa 1–308), and KCTD19-C (aa 694–950) were produced by inserting cDNA fragments in-frame with pET28c (Novagen) in *E. coli* strain BL21-CodonPlus(DE3)-RIPL (Agilent), solubilized in a denaturing buffer (6 M HCl-Guanidine, 20 mM Tris-HCl [pH 7.5]) and purified by Ni-NTA (QIAGEN) under denaturing conditions. Polyclonal antibodies against ZFP541-N were generated by

immunizing rabbits. Polyclonal antibodies against KCTD19-N were generated by immunizing rabbits, rats, and mice. Polyclonal antibodies against mouse KCTD19-C were generated by immunizing rabbits and rats. The antibodies were affinity-purified from the immunized serum with immobilized antigen peptides on CNBr-activated Sepharose (GE healthcare).

**Antibodies.** The following antibodies were used for immunoblot (IB) and immunofluorescence (IF) studies: rabbit anti-Actin (IB, 1:2000, sigma A2066), rabbit anti-SYCP1 (IF, 1:1000, Abcam ab15090), mouse anti-H2AX (IF, 1:1000, Abcam ab26350), rabbit anti-γH2AX (IF, 1:1000, Abcam ab11174), rabbit anti-H3S10p (IF, 1:2000, ab5176), rabbit anti-TDIF1 (IB, 1:1000, ab228703), rabbit anti-HDAC2 (IB, 1:1000, Abcam ab32117), mouse anti-HDAC1 (1:1000, Upstate 05-614), rabbit TDIF1 (IB, 1:1000, Abcam ab228703), guinea pig anti-H1t (IF, 1:2000, kindly provided by Marry Ann Handel[7], α-tubulin DM1A (IB, 1:2000, Sigma 05-829), rabbit anti-DAZL (IF, 1:1000, ab34139), mouse anti-SYCP1 (IF, 1:1000, our home made)[37], rat anti-SYCP3 (IF, 1:1000, our home made)[26], gunia pig anti-SYCP3 (IF, 1:2000, our home made)[26], mouse anti-SYCP3 (our home made)[37], rabbit anti-MEIKIN (our home made)[32] (IF, 1:1000), rat anti-STRA8(IF, 1:1000, our home made)[26], rabbit ZFP541-N (IF, IB, 1:1000), rabbit KCTD19-N (IB, 1:1000, our home made in this study), rat KCTD19-N (IF, 1:1000, our home made in this study), rabbit KCTD19-C (IF, 1:1000, our home made in this study), and rat KCTD19-C (IF, 1:1000, our home made in this study). Following secondary antibodies were used: Goat anti-rat IgG-Alexa Fluour 647 (IF, 1:1000, Thermo-Fisher, A21247), Goat anti-rabbit IgG-Alexa Fluour 647 (IF, 1:1000, ThermoFisher, A21244), Donkey anti-mouse IgG-Alexa Fluour 647 (IF, 1:1000, ThermoFisher, A31571), Donkey anti-rabbit IgG-Alexa Fluour 647 (IF, 1:1000, ThermoFisher, A31573), Donkey anti-rabbit IgG-Alexa Fluour 555 (IF, 1:1000, ThermoFisher, A31572), Donkey anti-rat IgG-Alexa Fluour 555 (IF, 1:1000, ThermoFisher, A48270), Donkey anti-rabbit IgG-Alexa Fluour 568 (IF, 1:1000, ThermoFisher, A10042), Goat anti-rat IgG-Alexa Fluour 568 (IF, 1:1000, ThermoFisher, A11077), Goat anti-rabbit IgG-Alexa Fluour 568 (IF, 1:1000, ThermoFisher, A11011), Goat anti-mouse IgG-Alexa Fluour 568 (IF, 1:1000, ThermoFisher, A11004), Donkey anti-rabbit IgG-Alexa Fluour 488 (IF, 1:1000, ThermoFisher, A21206), Donkey anti-rat IgG-Alexa Fluour 488 (IF, 1:1000, ThermoFisher, A21208), Goat anti-Gunia pig IgG-Alexa Fluour 488 (IF, 1:1000, Abcam ab150185), Goat anti-Gunia pig IgG-Alexa Fluour 555 (IF, 1:1000, Abcam ab150186), and Goat anti-Gunia pig IgG-Alexa Fluour 647 (IF, 1:1000, Abcam ab150187).

**Preparation of testis extracts and immunoprecipitation.** To prepare testis extracts, testes were removed from male C57BL/6 mice (P18-23), detunicated, and then resuspended in extraction buffer (20 mM Tris-HCl [pH 7.5], 100 mM KCl, 0.4 mM EDTA, 0.1% Triton X-100, 10% glycerol, 1 mM β-mercaptoethanol) supplemented with Complete Protease Inhibitor (Roche). After homogenization, the cell extracts were filtrated to remove debris. The soluble chromatin-unbound fraction was collected after ultra-centrifugation at $100,000 \times g$ for 30 min. The insoluble pellet was washed with buffer (10 mM Tris-HCl [pH 7.5], 1 mM CaCl$_2$, 1.5 mM MgCl$_2$, 10% glycerol) and digested with Micrococcus nuclease (0.008 units/ml) at 4 °C for 60 min. The solubilized fractions were removed after centrifugation at $20,000 \times g$ for 10 min at 4 °C. The chromatin-bound fractions were extracted from the insoluble pellet by high salt extraction buffer (20 mM HEPES-KOH [pH 7.0], 400 mM KCl, 5 mM MgCl$_2$, 0.1% Tween20, 10% glycerol, 1 mM β-mercaptoethanol) supplemented with Complete Protease Inhibitor. The solubilized chromatin fractions were collected after centrifugation at $100,000 \times g$ for 30 min at 4 °C.

For immunoprecipitation of endogenous ZFP541 and KCTD19 from extracts, 5 µg of affinity-purified rabbit anti-ZFP541, KCTD19-N, and control IgG antibodies were crosslinked to 50 µl of protein A-Dynabeads (ThermoFisher) by DMP (Sigma). The antibody-crosslinked beads were added to the testis extracts prepared from WT testes (P18). The beads were washed with a low salt extraction buffer. The bead-bound proteins were eluted with 40 µl of elution buffer (100 mM Glycine-HCl [pH 2.5], 150 mM NaCl), and then neutralized with 4 µl of 1 M Tris-HCl [pH 8.0]. The immunoprecipitated proteins were run on 4–12% NuPAGE (ThermoFisher) in MOPS-SDS buffer and immunoblotted. Immunoblot image was developed using ECL prime (GE Healthcare) and captured by LAS4000 (GE healthcare) or X-ray film.

**Mass spectrometry.** For mass spectrometry analysis of ZFP541 associated factors, fractions containing the ZFP541 immunoprecipitates were concentrated by precipitation with 10% trichloroacetic acid. The derived precipitates were dissolved in 7 M Urea, 50 mM Tris-HCl (pH 8.0), 5 mM EDTA solution, with 5 mM DTT at 37 °C for 30 min, and cysteine SH groups were alkylated with 10 mM iodoaceta-mide at 37 °C for 1 h. After alkylation, the solutions were desalted by methanol/chloroform precipitation, and the precipitates were dissolved in 2 M urea, 50 mM Tris-HCl buffer and subjected to trypsin gold (Promega) digestion overnight at 37 °C. The resulting mixture of peptides was applied directly to the LC–MS/MS analysis system (Zaplous, AMR, Tokyo, Japan) using Finnigan LTQ mass spectrometry (Thermo Scientific) and a reverse phase C18 ESI column (0.2 × 50 mm, LC assist). The protein annotation data were verified in the mouse NCBI sequences

using Bioworks software (Ver. 3.3; Thermo Scientific) with quantitation featuring the SEQUEST search algorithm. Two independent samples of ZFP541-IP from chromatin bound-, chromatin unbound-, MNase solubilized nucleosome fractions were analyzed. Identified proteins were presented after excluding the proteins detected in the control IgG-IP.

For mass spectrometry analysis of KCTD19-IP, the immunoprecipitated proteins were run on 4–12% NuPAGE (ThermoFisher Scientific) by 1 cm from the well and stained with SimplyBlue (LC6065, ThermoFisher Scientific) for the in-gel digestion. The gel that contained proteins was excised, cut into ~1 mm sized pieces. Proteins in the gel pieces were reduced with DTT (20291, ThermoFisher Scientific), alkylated with iodoacetamide (90034, ThermoFisher Scientific), and digested with trypsin and lysyl endopeptidase (Promega, USA) in a buffer containing 40 mM ammonium bicarbonate, pH 8.0, overnight at 37 °C. The resultant peptides were analyzed on an Advance UHPLC system (AMR/Michrom Bioresource) coupled to a Q Exactive mass spectrometer (ThermoFisher Scientific) processing the raw mass spectrum using Xcalibur version 4.0 (ThermoFisher Scientific). The raw LC–MS/MS data were analyzed against the NCBI non-redundant protein and UniProt restricted to *Mus musculus* using Proteome Discoverer version 1.4 (ThermoFisher Scientific) with the Mascot search engine version 2.5 (Matrix Science). A decoy database comprised of either randomized or reversed sequences in the target database was used for false discovery rate (FDR) estimation, and Percolator algorithm was used to evaluate false positives. Search results were filtered against 1% global FDR for high confidence level. Identified proteins were presented after excluding the proteins detected in the control IgG-IP.

**Histological analysis.** For hematoxylin and eosin staining, testes, epididymis, and ovaries were fixed in 10% formalin or Bouin solution, and embedded in paraffin. Sections were prepared on MAS-GP typeA-coated slides (Matsunami) at 6 µm thickness. The slides were dehydrated and stained with hematoxylin and eosin.

For Immunofluorescence staining, testes and embryonic ovaries were embedded in Tissue-Tek O.C.T. compound (Sakura Finetek) and frozen. Cryosections were prepared on the MAS-GP typeA-coated slides (Matsunami) at 8 µm thickness, and then air-dried and fixed in 4% paraformaldehyde in PBS at pH 7.4. The serial sections of frozen testes were fixed in 4% PFA for 5 min at room temperature and permeabilized in 0.1% Triton X-100 in PBS for 10 min. The sections were blocked in 3% BSA/PBS or Blocking One (Nakarai), and incubated at room temperature with the primary antibodies in a blocking solution. After three washes in PBS, the sections were incubated for 1 h at room temperature with Alexa-dye-conjugated secondary antibodies (1:1000; Invitrogen) in a blocking solution. TUNEL assay was performed using MEBSTAIN Apoptosis TUNEL Kit Direct (MBL 8445). PNA lectin staining was done using FITC-conjugated Lectin from *Arachis hypogaea* (IF, 1:1000, Sigma: L7381). DNA was counterstained with Vectashield mounting medium containing DAPI (Vector Laboratory). Statistical analyses, and production of graphs and plots were done using GraphPad Prism8 (version 8.4.3) or Microsoft Excel (version 16.48).

**Immunostaining of spermatocytes.** Spread nuclei from spermatocytes were prepared as described previously[43]. Briefly, testicular cells were suspended in PBS, then dropped onto a slide glass together with an equal volume of 2% PFA, 0.2% (v/v) Triton X-100 in PBS, and incubated at room temperature in a humidified chamber. The slides were then air-dried and washed with PBS containing 0.1% Triton X-100 or frozen for longer storage at −80 °C. The slides were immunostained as described above.

**Fluorescence in situ hybridization (FISH) on immunostained nuclei.** For immuno-FISH, spread nuclei from spermatocytes were prepared as described[43]. Immunostained samples of spread nuclei were fixed in 4% paraformaldehyde for 5 min, washed with PBS, and subjected to FISH. Immunostained nuclei were denatured in 50% formamide, 2× SSC at 72 °C for 10 min. Hybridization was conducted with a biotinylated point probe that detects X chromosome and a FITC-labeled painting probe that detects the whole Y chromosome (MXY-10, Chromosome Science Labo, Hokkaido, Japan) in buffer containing 50% formamide, 2× SSC, 20% dextran sulfate at 37 °C for 12–16 h. The slides were washed sequentially at room temperature in 2× SSC for 1 min, 0.4× SSC/0.3% Tween20 solution for 2 min, and 2× SSC at room temperature for 1 min. The biotinylated X chromosome point probe was detected by Alexa555-streptavidin (Thermo, S21381).

**Imaging.** Immunostaining images were captured with DeltaVision (GE Healthcare). The projection of the images was processed with the SoftWorx software program (version 7.2.1, GE Healthcare). All images shown were Z-stacked. Brightfield images were captured with OLYMPUS BX53 fluorescence microscope and processed with CellSens standard program.

**Giemsa staining of metaphase I chromosome spread.** The spermatocytes were treated in the hypotonic buffer for 10 min and fixed in the Carnoy's Fixative (75% methanol, 25% acetic acid) and, stained in 3% Giemsa solution for 30 min.

**Chromatin immunoprecipitation**. The seminiferous tubules from male C57BL/6 mice (P18-19-old age) were minced and digested with accutase and 0.5 units/ml DNase II, followed by filtration through a 40 μm cell strainer (FALCON). Testicular cells were fixed in 1% formaldehyde (ThermoFisher)/2 mM Disuccinimidyl glutarate (ProteoChem) in PBS for 10 minutes at room temperature. Crosslinked cells were lysed with LB1 (50 mM HEPES pH 7.5, 140 mM NaCl, 1 mM EDTA, 10% glycerol, 0.5% NP-40, and 0.25% Triton X-100) and washed with LB2 (10 mM Tris-HCl pH 8.0, 200 mM NaCl, 1 mM EDTA, 0.5 mM EGTA). Chromatin lysates were prepared in LB3 (50 mM Tris-HCl pH 8.0, 1% SDS, 10 mM EDTA, proteinase inhibitor cocktail (Sigma)), by sonication with Covaris S220 (Peak Incident Power, 175; Acoustic Duty Factor, 10%; Cycle Per Burst, 200; Treatment time, 600 s; Cycle, 2).

ZFP541 ChIP was performed using chromatin lysates and protein A Dynabeads (ThermoFisher) coupled with 5 μg of rabbit anti-ZFP541-N antibodies (#1 and #2). After 4 h of incubation at 4 °C, beads were washed four times in a low salt buffer (20 mM Tris-HCl (pH 8.0), 0.1% SDS, 1% (w/v) Triton X-100, 2 mM EDTA, 150 mM NaCl), and two times with a high salt buffer (20 mM Tris-HCl (pH 8.0), 0.1% SDS, 1% (w/v) Triton X-100, 2 mM EDTA, 500 mM NaCl). Chromatin complexes were eluted from the beads by agitation in elution buffer (10 mM Tris-HCl (pH 8.0), 300 mM NaCl, 5 mM EDTA, 1% SDS) and incubated overnight at 65 °C for reverse-crosslinking. Eluates were treated with RNase A and Proteinase K, and DNA was ethanol precipitated.

**ChIP-seq data analysis**. ChIP-seq libraries were prepared using 20 ng of input DNA, and 4 ng of ZFP541-ChIP DNA with KAPA Library Preparation Kit (KAPA Biosystems) and NimbleGen SeqCap Adaptor Kit A or B (Roche) and sequenced by Illumina Hiseq 1500 to obtain single-end 50 nt reads.

ChIP-seq reads were trimmed to remove adapter sequence when converting to a fastq file. The trimmed ChIP-seq reads were mapped to the UCSC mm10 genome assemblies using Bowtie2 v2.3.4.1 with default parameters. Peak calling was performed using MACS program v2.1.0[44] (https://github.com/macs3-project/MACS) with the option (-g mm -p 0.00001). To calculate the number of overlapping peaks between Ab-1 and Ab-2, we used bedtools program (v2.27.1)[45] (https://bedtools.readthedocs.io/en/latest/). ChIP binding regions were annotated with BED file using Cis-regulatory Element System (CEAS) v0.9.9.7 (package version 1.0.2), in which the gene annotation table was derived from UCSC mm10. Motif identification was performed using MEME-ChIP v5.1.1 website (http://meme-suite.org/tools/meme-chip)[46]. The motif database chosen was "JASPAR Vertebrates and UniPROBE Mouse". BigWig files, which indicate occupancy of ZFP541, were generated using deepTools (v3.1.0) and visualized with Integrative Genomics Viewer software (v.2.8.3) http://software.broadinstitute.org/software/igv/home. GO-term analyses were performed using DAVID Bioinformatics Resources 6.8[47] (https://david.ncifcrf.gov/). Aggregation plots and heatmaps for each sample against the data set of ZFP541 target genes were made using deeptools program v3.5.0. The nearest genes of the ChIP-seq peaks were determined by GREAT website (v4.0.4) (http://great.stanford.edu/public/html/). Enrichment analyses of H3K27me3 and H3K27ac over ZFP541-bound targets were performed with bedtools program (v2.27.1) with -c option.

**CAGE-seq**. CAGE data were downloaded from GEO with accession number GSE44690. Downloaded fastq file was trimmed to remove adapter sequence and low-quality ends using Trim Galore! v0.4.3 (cutadapt v1.15). The trimmed reads ware mapped on the mouse genome GRCm38.92 using TopHat (v2.1.1), using GTF file (-G (—GTF) GRCm38.92) with option of -g (—max-multihits) 1.

**RNA-seq data analysis**. GFP-positive cells from *Rec8-3xFLAG-HA-p2A-GFP* knock-in male testis were sorted using SH800S cell sorter (SONY). Total RNAs were prepared by Trizol (ThermoFisher) and the quality of total RNA was confirmed by BioAnalyzer 2100 (RIN > 8) (Agilent). Library DNAs were prepared according to the Illumina Truseq protocol using Truseq Standard mRNA LT Sample Prep Kit (Illumina) and sequenced by Illumina NextSeq 500 (Illumina) using Nextseq 500/550 High Output v2.5 Kit (Illumina) to obtain single-end 75 nt reads.

Resulting reads were aligned to the mouse genome UCSC mm10 using STAR ver.2.6.0a after trimmed to remove adapter sequence and low-quality ends using Trim Galore! v0.5.0 (cutadapt v1.16). Gene expression level measured as RPKM was determined by Cuffdiff v 2.2.1. Differential expression analysis using TPM was done by RSEM v1.3.1. GTF file was derived from UCSC mm10. Principal component analysis and GO-term analysis were performed using DAVID Bioinformatics Resources 6.8[47].

**Single-cell RNA-seq data analysis**. The scRNA-seq data of fetal ovaries were derived from DRA 011172[31]. 10xGenomics Drop-seq data of mouse adult testis were derived from GEO: GSE109033(Hermann et al., 2018). Reanalyses of scRNA-seq data were conducted using the Seurat package for R (v.3.1.3)[48] and pseudotime analyses were conducted using monocle package for R; R (ver. 3.6.2), RStudio (ver.1.2.1335), and monocle (ver. 2.14.0)[49,50] following developer's tutorial.

**Reporting summary**. Further information on research design is available in the Nature Research Reporting Summary linked to this article.

## Data availability

All data supporting the conclusions are present in the paper and the supplementary materials. A reporting summary for this Article is available as a Supplementary Information file. Uncropped blots can be found in Supplementary Fig. 11. The original images for all of the figures in this paper are deposited in a public depository https://doi.org/10.17632/kxb38h7snx.3. The ZFP541 ChIP-seq data of mouse testes have been deposited in the GEO Sequence Read Archive (SRA) database under accession code: GSE163916. RNA-seq data of the GFP-positive cells from the control and *Zfp541* KO have been deposited in the GEO Sequence Read Archive (SRA) database under accession code: GSE163917. ChIP-seq data of H3K27me3[24] and H3K27ac[14] were derived from GSE89502 and GSE130652, respectively. Data for RNA-seq (THY1 + SG, PS, RS)[22] and RNA-seq (KIT + SG)[24] were derived from GSE55060 and GSE89502, respectively. CAGE data were derived from GEO: GSE44690[38]. The scRNA-seq data of fetal ovaries was derived from DRA 011172[31]. 10×Genomics Drop-seq data of mouse adult testis was derived from GEO: GSE109033[29]. Source data are provided with this paper.

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

## Acknowledgements

We thank Drs. Masahito Ikawa, Seiya Oura (Osaka University) for discussion and sharing unpublished data; Sally Fujiyama (The University of Tokyo), Etsushi Kitamura, Kaho Okamura (Kumamoto University) for technical support, and Marry Ann Handel for provision of H1t antibody. This work was supported in part by the program of the Joint Usage/Research Center for Developmental Medicine, IMEG, Kumamoto University (to S.N.), KAKENHI grant (#19K06642); Takeda Science Foundation (to Y.T.), KAKENHI grant (#20K22638) (to R.S.), and KAKENHI grants (#17H03634, #18K19304, #19H05245, #19H05743, #20H03265, #20K21504, #JP 16H06276) from MEXT Japan; Joint Usage and Joint Research Programs from the Institute of Advanced Medical Sciences, Tokushima University (30-A-7); Grants from The Sumitomo Foundation; The Naito Foundation, Astellas Foundation for Research on Metabolic Disorders; Daiichi Sankyo Foundation of Life Science; The Uehara Memorial Foundation; The NOVARTIS Foundation (Japan) for the promotion of Science (to K.-i.I).

## Author contributions

Y.-H.T., C.K., K.T., and R.S. performed the mice cytology experiments. K.T., K.-i.I, and N.T. performed MS analyses. Y.-H.T. performed the ChIP-seq, supported by K.M., H.N., R.M., M.T., and T.A. Y.T. performed the RNA-seq, supported by S.U. K.M., K.H., and A.Suzuki supported data analyses of ChIP-seq and RNA-seq. A.Sakashita and S.N. performed a reanalysis of RNA-seq and ChIP-seq data. R.S. performed a reanalysis of scRNA-seq. S.F. performed histological analyses. K.A. and K.-i.I. designed the knockout mice. K.-i.I. supervised, conducted the study, and wrote the manuscript.

## Competing interests

The authors declare no competing interests.
