## [Peer Review File · Nature Communications]

Reviewers' Comments:

Reviewer #1:

Remarks to the Author:

Overview: In this manuscript the authors report on the identification of a new transcriptional regulating complex that is required in mammalian spermatocytes for normal progress to the conclusion of meiotic prophase and transition to the division phase and spermiogenic differentiation. The findings are significant as they could lead to identification of the downstream drivers of these crucial steps in spermatogenesis.

For the most part, the data are solid and support the overall conclusions. The manuscript follows a logical flow and is clearly written. The figures are mostly of good quality with clear legends (but see comments below). Overall, the results are of considerable interest to an audience interested in meiosis and gametogenesis, with broader relevance to transcriptional regulatory pathways in general. The authors could enhance the impact and message of this report by considering the following comments and suggestions:

Seminiferous tubule sections depicted (especially in Fig. 1, but also Fig. 2, especially 2J, and Fig. 3) are a bit difficult to resolve and could be improved with staging and more attention to selection of informative tubules. In part this is because boundaries of the tubules cannot easily be visualized; addition of white dashed lines delimiting the tubules would help immensely. Additionally, there would be great benefit from incorporating more information about the relationship of ZFP541-expressing cells and mutant phenotypes to the context of the stages of the seminiferous epithelium. First, the seminiferous tubule sections would be more easily interpreted and informative if staged. But more importantly, identifying stage specificity of expression would be highly advantageous because of the extra information that might be derived, and would considerably enhance the model presented in Fig. 7 (where substage timing is vague). For example, it is known when androgen signaling peaks during the cycle of the seminiferous epithelium (stage VI-VII); is there any relationship to the peak expression of germ-cell ZFP541? We know that that AR-dependent signaling from Sertoli cells is required for exit of spermatocytes from prophase and entry to the division phase. Thus defining ZFP541 expression patterns in terms not just of the germ cell but of the stages of the seminiferous epithelium could allow these features to be related. For instance, is there any evidence that the Zfp541 gene might be regulated by the cascade of germ-cell gene expression downstream of Sertoli-cell AR-dependent signaling to germ cells? Even if the authors cannot comment directly on this point, information on its stage specificity would enhance biological understanding of the role of ZFP541 and contribute to the growing body of literature that is developing a more "systems" view of dynamics of spermatogenesis.

The authors should seriously reconsider their language use in referring to ZFP541 as regulating meiotic prophase "exit" because it is probably more accurate to state that it regulates prophase "progression." They have clearly shown a role for ZFP541 in transcriptional repression, contributing to reconstructing a pattern of gene expression; this process is an essential element in the normal "progression" of meiotic prophase. However, in the view of many, meiotic "exit" per se refers specifically to the events at the pachytene-to-diplotene stage transition. Meiotic "exit" defined in this way involves MPF activation and a role for HSPA2. Meiotic exit can be prematurely promoted by phosphatase inhibitors in those spermatocytes that have accumulated histone H1t, although H1t is not required for meiotic exit. It is not known what factors contribute to enabling the cell-cycle events that constitute meiotic exit, except that, as mentioned above, it requires androgen signaling from Sertoli cells. The authors should give some attention to these considerations and the supporting literature. Although they have provided clear evidence that the Zfp541 mutant phenotype is arrest prior to the meiotic division phase, there is no direct evidence that this is solely because of failure of proximal cell-cycle regulators of this diplotene transition. Many genes appear to be normally repressed by ZFP541 and it is not yet known early repression of any of these could prematurely promote the exit from prophase as biologically defined above. Thus, with

the available evidence on the mutant Zfp541 phenotype, it would be more accurate to state that the ZFP541 repressor complex promotes developmental "progression" of meiotic prophase toward completion, rather than specifically regulating meiotic "exit."

Other Comments:

Line 129: because the staining is not robust, it is hard to evaluate the timing of ZFP541 and if this should refer to Fig. 1E or 1F

Line 154-155: SYCP3 was defined above, but SYCP1 function should be defined here

Line 242 and Fig. 4J: left panel not of good quality; the mutant is a much better meiotic prep!

Fig. S5: although the characterization of the mutant Rec8-GFP knock-in is good, what is really needed for the transcriptome analysis is information of the distribution of meiotic substages among the sorted cells used for RNA-seq – this is crucial for interpreting the results on up- and down-regulated transcripts in the mutant

Lines 384-385: delete "in" before "downstream"

Summary: This is a very interesting manuscript reflecting excellence in the conduct of the investigation. It can be enhanced and made more valuable by 1) greater precision in defining the limitations on what is known about progression and tempo of meiotic prophase and specific "exit" from prophase via the pachytene-to-diplotene transition, and 2) improving figures for clarity and to address the findings in the context of the cycle of the seminiferous epithelium.

Reviewer #2:

Remarks to the Author:

This manuscript from Horisawa-Takada and colleagues reports discovery of ZFP541 as a factor required for extinguishing the meiotic program in spermatocytes. This builds upon prior results demonstrating activation of this program by MEIOSIN and STRA8. The promoter of the gene encoding ZFP541 appears to be bound by MEIOSIN/STRA8, but this is not confirmed by knockouts. The authors claim ZFP541 is expressed in pachytene spermatocytes through round spermatids, but the analysis is superficial and the authors ignore the role of stage of the cycle of the seminiferous epithelium. Proving expressing cell identity should involve chromosome spreads and costaining with cell type specific markers (which was done only to a very limited extent, SYCP3 and H1t). In the ovary, STRA8 expression occurs in an anterior \diamond posterior wave, and thus, simple immunostaining while ignoring spatial context fails to support regulation by STRA8. Including existing single-cell data would likewise be helpful in establishing expression profile in the testis and ovary. Zfp541 KOs appear to have a spermatogenic block in pachynema and the phenotype is mostly well characterized (could use better tools to resolve cell types, such as FISH), but there is a bimodal distribution of MLH1 foci that is never explored. The authors claim these mice are infertile, but never show breeding data and limited information is presented about spermatids present. KCTD19 is identified as a ZFP541-interacting protein through IP-mass spec and show a similar expression profile (although also similarly superficial in depth of expression profile analysis). Again, including more thorough analysis of expression by stage, with chromosome spreads and depiction of single-cell results would be helpful. Kctd19 KOs also have a meiotic block phenotype, but somewhat delayed and only partially phenocopy the Zfp541 KOs, indicating some KCTD19-independent functions of ZFP541. To explore the target genes of ZFP541, the authors devise a new model to select spermatocytes with a new transgenic Rec8-GFP, but it is not clear why such a new model is needed given the availability of other methods to isolate spermatocytes. Bulk RNA-seq data from P18 testes (which are known to be heterogeneous in spermatocyte development) showed DEGs and the authors bin DEGs by whether those genes are expressed in Pachytene Sct, Round Std, THY1+ (Aundiff) or KIT+ (Diff), notably – neither of these "spermatogonial" genes is germ cell specific – in an attempt to correlate gene expression regulation with developmental progression. These results appear to show batch effects that are linked to differences in cell proportions, so it is unclear why single-cell was not used to provide meaningful results. ChIP-seq using two novel ZFP541 antibodies predicted genomic binding sites in spermatocytes, but

surprisingly, the simplest negative control – Zfp541KOs was not included – a very surprising oversight – and a very small proportion of the binding sites are associated with change in gene expression (and most changes in gene expression are not associated with a binding event). This emphasizes the two key concerns that the DEG analysis (bulk RNA-seq) was not done with an appropriate sample type and ChIP was not properly controlled.

In general, I found the data to be of very high quality, the experiments thoughtful and elegantly presented, but as noted above, the conclusions were not always supported by the results. Thus, there is significant room for improvement to better align the conclusions with the extent of results and provide meaningful molecular results in Figs. 5-6 to overcome technical problems. If the concerns can be addressed, then I believe this will represent a very valuable contribution to the field of spermatogenesis by revealing the requisite molecular players is extinguishing the meiotic program prior to spermiogenesis.

The following specific criticisms should also be addressed:

1. A number of typographical errors were noted throughout the manuscript which should be addressed upon revision.
2. Fig. 1A – Negative control ChIP with Stra8 KOs should be shown.
3. Fig. 1B – expression in Stra8 KOs should be shown.
4. Fig. 1E – this panel does not adequately assess expression of ZFP541 and should be expanded to show each stage of the cycle of the seminiferous epithelium with cell type markers and chromosome spreads (like in Fig. 2G) to confirm cell types. PNA co-labeling would be useful for identifying spermatid subtypes by stage to support claims of absence in elongating spermatids.
5. Fig. 2G – the sex body in KOs hardly looks normal – need to confirm this with DNA FISH against the sex chromosomes.
6. Fig. 2J – these data are insufficient to draw conclusions about MI – need metaphase spreads.
7. Fig. 2K – TUNEL% of SYCP3+ vs. PNA+ would be much more informative than per tubule. Also, how were stages defined (authors claimed stages X-XI tubules, but no evidence provided)? Why are only <50% tubules affected?
8. Lines 184-185 – this sentence is not sufficiently supported by data.
9. Fig. S2 – beautiful data!
10. Fig. 3C-E should be expanded to consider stage of the epithelial cycle.
11. Fig. 4C – not sufficient to conclude nuclear localization.
12. Fig. 4H – the bimodal distribution of MLH1 foci in KOs is very interesting – what is the explanation?
13. Fig. 4K – This should also be done in Zfp541 KOs.
14. Fig. 4M – same comments as Fig. 2K.
15. Fig. 5 – neither THY1 nor KIT are unique to spermatogonia, so these gene sets do not bin to Spermatogonia, as claimed. Suggest using single-cell datasets to provide a more robust and higher resolution separation of genes changed in the KO.
16. Fig. 6D genes (lines 317-320) – the mRNA expression profiles of the noted genes are highly complex across spermatogenic progression, and thus, their direct regulation by ZFP541 is not easy to establish – rather than oversimplifying, the authors should be more carefully present these as targets and show the relevant expression profiles.
17. Fig. 6 – validation/confirmation by ChIP-qPCR is lacking.
18. Fig. 7 – this model figure is highly speculative, especially when it comes to histone modification changes. This figure should be revised to only cover results supported by data at hand.

REVIEWER COMMENTS

Reviewer #1 (Remarks to the Author):

Overview: In this manuscript the authors report on the identification of a new transcriptional regulating complex that is required in mammalian spermatocytes for normal progress to the conclusion of meiotic prophase and transition to the division phase and spermiogenic differentiation. The findings are significant as they could lead to identification of the downstream drivers of these crucial steps in spermatogenesis.

For the most part, the data are solid and support the overall conclusions. The manuscript follows a logical flow and is clearly written. The figures are mostly of good quality with clear legends (but see comments below). Overall, the results are of considerable interest to an audience interested in meiosis and gametogenesis, with broader relevance to transcriptional regulatory pathways in general. The authors could enhance the impact and message of this report by considering the following comments and suggestions:

Seminiferous tubule sections depicted (especially in Fig. 1, but also Fig. 2, especially 2J, and Fig. 3) are a bit difficult to resolve and could be improved with staging and more attention to selection of informative tubules. In part this is because boundaries of the tubules cannot easily be visualized; addition of white dashed lines delimiting the tubules would help immensely. Additionally, there would be great benefit from incorporating more information about the relationship of ZFP541-expressing cells and mutant phenotypes to the context of the stages of the seminiferous epithelium. First, the seminiferous tubule sections would be more easily interpreted and informative if staged.

But more importantly, identifying stage specificity of expression would be highly advantageous because of the extra information that might be derived, and would considerably enhance the model presented in Fig. 7 (where substage timing is vague). For example, it is known when androgen signaling peaks during the cycle of the seminiferous epithelium (stage VI-VII); is there any relationship to the peak expression of germ-cell ZFP541? We know that that AR-dependent signaling from Sertoli cells is required for exit of spermatocytes from prophase and entry to the division phase. Thus defining ZFP541 expression patterns in terms not just of the germ cell but of the stages of the seminiferous epithelium could allow these features to be related. For instance, is there any evidence that the Zfp541 gene might be regulated by the cascade of germ-cell gene expression downstream of Sertoli-cell AR-dependent signaling to germ cells? Even if the authors cannot comment directly on this point, information on its stage specificity would enhance biological understanding of the role of ZFP541 and contribute to the growing body of literature that is developing a more “systems” view of dynamics of spermatogenesis.

We appreciate the reviewer’s thoughtful suggestion to improve our manuscript.

We newly presented immunostaining of spread chromosomes, showing ZFP541 and KCTD19 expressions in meiotic prophase stages and round spermatids in the new Fig1e and Fig3c. We examined ZFP541 and KCTD19 expressions by immunostaining of seminiferous tubules using PNA and SYCP3 in the new supplementary Fig2, and supplementary Fig7, Fig1f, Fig2f,k, I, and Fig3e. We confirmed that ZFP541 signals started to appear in the nuclei of pachytene spermatocytes of stage I seminiferous tubules and in round spermatids but were absent in elongated spermatids.

We further examined seminiferous tubules in WT vs *Zfp541*KO by additional immunostaining. The existence of pachytene spermatocytes up to stage IX was comparable between WT and *Zfp541* KO (new supplementary Fig4), and the stage X tubules of *Zfp541* KO became TUNEL positive. We mentioned these observations in the main text (Line 170 – 171, Line 200-203, Line 210-222).

Accordingly, the schematic model was revised in the new Fig. 7.

Boundaries of the seminiferous tubules are now indicated by white dashed lines. If the stages can be unambiguously assigned, seminiferous stages are indicated in the immunostaining images of seminiferous tubules throughout the figures.

We did not find relationship between the peak expression of ZFP541 in germ cells and the AR-dependent signaling from Sertoli that peaks at stage VI-VII during the cycle of the seminiferous epithelium (Larose et al., 2020). It is possible that the extrinsic response of the germ cells to androgen signaling by Sertoli cells may coordinately function with the intrinsic gene expressions of the germ cells to promote meiotic prophase completion and the subsequent first division phase. We added this in the discussion (Line 440- 450).

The authors should seriously reconsider their language use in referring to ZFP541 as regulating meiotic prophase “exit” because it is probably more accurate to state that it regulates prophase “progression.” They have clearly shown a role for ZFP541 in transcriptional repression, contributing to reconstructing a pattern of gene expression; this process is an essential element in the normal “progression” of meiotic prophase. However, in the view of many, meiotic “exit” per se refers specifically to the events at the pachytene-to-diplotene stage transition. Meiotic “exit” defined in this way involves MPF activation and a role for HSPA2. Meiotic exit can be prematurely promoted by phosphatase inhibitors in those spermatocytes that have accumulated histone H1t, although H1t is not required for meiotic exit. It is not known what factors contribute to enabling the cell-cycle events that constitute meiotic exit, except that, as mentioned above, it requires androgen signaling from Sertoli cells. The authors should give some attention to these considerations and the supporting literature.

In the introduction, we added sentences and previous literatures that are related to the events at the pachytene-to-diplotene stage transition (Line 60-62, 74-77). In addition to germ-cell intrinsic gene expression, we mentioned that germ-cell extrinsic genes responsive to androgen signaling in Sertoli cells promote cellular states permissive for meiotic prophase completion and subsequently the first division phase (Larose et al., 2020) in introduction and discussion.

Although they have provided clear evidence that the *Zfp541* mutant phenotype is arrest prior to the meiotic division phase, there is no direct evidence that this is solely because of failure of proximal cell-cycle regulators of this diplotene transition. Many genes appear to be normally repressed by ZFP541 and it is not yet known early repression of any of these could prematurely promote the exit from prophase as biologically defined above. Thus, with the available evidence on the mutant *Zfp541* phenotype, it would be more accurate to state that the ZFP541 repressor

complex promotes developmental “progression” of meiotic prophase toward completion, rather than specifically regulating meiotic “exit.”

We appreciate the reviewer’s suggestion. We changed the title to “Meiosis-specific ZFP541 repressor complex promotes developmental progression of meiotic prophase toward completion during spermatogenesis”. Accordingly, in the main text, we rephrased the related sentences so that ZFP541 repressor complex promotes “progression” of meiotic prophase, but not “exit”.

Other Comments:

Line 129: because the staining is not robust, it is hard to evaluate the timing of ZFP541 and if this should refer to Fig. 1E or 1F

In the new Fig1E, we presented immunostaining of spread chromosome, showing ZFP541 expression from pachytene to round spermatids.

Line 154-155: SYCP3 was defined above, but SYCP1 function should be defined here
SYCP1 is now defined. Line 182.

Line 242 and Fig. 4J: left panel not of good quality; the mutant is a much better meiotic prep!
The Giemsa staining image of WT bivalents is replaced with a better one.

Fig. S5: although the characterization of the mutant Rec8-GFP knock-in is good, what is really needed for the transcriptome analysis is information of the distribution of meiotic substages among the sorted cells used for RNA-seq – this is crucial for interpreting the results on up- and down-regulated transcripts in the mutant

GFP positive population isolated from *Rec8-3FH-GFP* KI background at P18 contained ~ 50% of H1t positive pachytene cells among the prophase cells. This information is now presented in the new Supplementary Fig 9f. Assuming from the data of meiotic prophase composition at P21(Fig2g), we reasoned that GFP positive population contained a similar cellular composition in WT and *Zfp541* KO at P18, since *Zfp541* KO spermatocytes showed meiotic defects at late pachytene but before diplotene. Thus, a potential batch effect due to the bias in cellular population was minimized for the RNA-seq data. We added the related sentences in the new manuscript (Line 339-344).

Lines 384-385: delete “in” before “downstream”

We revised this.

Summary: This is a very interesting manuscript reflecting excellence in the conduct of the investigation. It can be enhanced and made more valuable by 1) greater precision in defining the limitations on what is known about progression and tempo of meiotic prophase and specific “exit” from prophase via the pachytene-to-diplotene transition, and 2) improving figures for clarity and to address the findings in the context of the cycle of the seminiferous epithelium.

Reviewer #2 (Remarks to the Author):

This manuscript from Horisawa-Takada and colleagues reports discovery of ZFP541 as a factor required for extinguishing the meiotic program in spermatocytes. This builds upon prior results demonstrating activation of this program by MEIOSIN and STRA8. The promoter of the gene encoding ZFP541 appears to be bound by MEIOSIN/STRA8, but this is not confirmed by knockouts. The authors claim ZFP541 is expressed in pachytene spermatocytes through round spermatids, but the analysis is superficial and the authors ignore the role of stage of the cycle of the seminiferous epithelium. Proving expressing cell identity should involve chromosome spreads and costaining with cell type specific markers (which was done only to a very limited extent, SYCP3 and H1t).

In the new Fig1E, we presented immunostaining of spread chromosome using SYCP1 and SYCP3 markers, showing ZFP541 expression at pachytene onward in meiotic prophase spermatocytes and in round spermatids.

Furthermore, we examined ZFP541 expression by immunostaining of the seminiferous tubules together with other cell type specific markers PNA, SYCP1, MEIKIN, PLZF, STRA8. We confirmed that ZFP541 signal started to appear faintly in early pachytene spermatocyte nuclei in stage I seminiferous tubules, and persisted in round spermatids in stage VII seminiferous tubules, but not in elongating spermatids (new supplementary Fig2a).

We also mentioned these in the revised main text (Line137-140).

In the ovary, STRA8 expression occurs in an anterior \diamond posterior wave, and thus, simple immunostaining while ignoring spatial context fails to support regulation by STRA8.

Including existing single-cell data would likewise be helpful in establishing expression profile in the testis and ovary.

We performed reanalysis of the existing scRNA-seq data of fetal ovaries (DRA 011172) which was previously published by one of the coauthors, Ryuki Shimada (Shimada et al, bioRxiv 2020, currently iScience under revision). Pseudotime analysis indicates that *Zfp541* and *Stra8* are co-ordinately upregulated along the pseudotime of fetal germ cell development .

Now, we presented expression profiles of *Zfp541* and *Stra8* along pseudotime in the new Supplementary Fig3b. Accordingly, we added the sentence (Line 152-157) “This observation was confirmed by the reanalysis of previous scRNA-seq data of fetal ovaries (Shimada et al, bioRxiv 2020). We found that the gene expression of *Zfp541* was coordinately upregulated with that of *Stra8* along the pseudotime of fetal oocyte development,.....”

Currently, we are trying to identify the gene sets regulated by MEIOSIN/STRA8 in female. At this moment, we still do not have the evidence that *Zfp541* is one of the target genes by MEIOSIN/STRA8 in female as shown in male. Thus, we added the sentence “although we had not specified yet whether MEIOSIN-STRA8 complex was involved in the ovarian *Zfp541* expression as in the spermatocytes.” (Line 156-157)

Zfp541 KOs appear to have a spermatogenic block in pachynema and the phenotype is mostly well characterized (could use better tools to resolve cell types, such as FISH), but there is a bimodal distribution of MLH1 foci that is never explored. The authors claim these mice are infertile, but never show breeding data and limited information is presented about spermatids present.

We showed that sperms were absent in adult *Zfp541* KO epididymis (Fig 2D), and round spermatids and elongated spermatids were absent in adult *Zfp541* KO seminiferous tubules (Fig 2D).

As we did not show breeding data of *Zfp541* knockout males, we rephrased the sentence “*Zfp541* knockout male is infertile” to “spermatogenesis was impaired in *Zfp541* knockout male” (Fig 2 caption).

KCTD19 is identified as a ZFP541-interacting protein through IP-mass spec and show a similar expression profile (although also similarly superficial in depth of expression profile analysis). Again, including more thorough analysis of expression by stage, with chromosome spreads and depiction of single-cell results would be helpful.

In the new Fig3C, we presented immunostaining of spread chromosome, showing the stages of KCTD19 expression in meiotic prophase spermatocyte and round spermatid.

We presented spermatogenic expression of *Zfp541* and *Kctd19* genes by reanalyzing previous scRNA-seq data (GSE109033) of adult mouse testis (Hermann et al., 2018). The result indicated that *Zfp541* and *Kctd19* were coordinately expressed with landmark genes of meiosis such as *Spo11* rather than those of spermiogenesis (*Prm1*) and spermatogonia (*Zbtb16*) along the pseudotime of spermatogenic development (Supplementary Fig. 1). We described this in the main text (Line 256-262).

Kctd19 KOs also have a meiotic block phenotype, but somewhat delayed and only partially phenocopy the *Zfp541* KOs, indicating some KCTD19-independent functions of ZFP541. To explore the target genes of ZFP541, the authors devise a new model to select spermatocytes with a new transgenic *Rec8-GFP*, but it is not clear why such a new model is needed given the availability of other methods to isolate spermatocytes. Bulk RNA-seq data from P18 testes (which are known to be heterogeneous in spermatocyte development) showed DEGs and the authors bin DEGs by whether those genes are expressed in Pachytene Sct, Round Std, THY1+ (Aundiff) or KIT+ (Diff), notably – neither of these “spermatogonial” genes is germ cell specific – in an attempt to correlate gene expression regulation with developmental progression. These results appear to show batch effects that are linked to differences in cell proportions, so it is unclear why single-cell was not used to provide meaningful results.

In our RNA-seq analysis, GFP positive population isolated from *Rec8-3FH-GFP* KI background at P18 contained ~ 50% of H1t positive pachytene cells among the prophase cells. Assuming from the data of meiotic prophase composition at P21 (Fig2g), we reasoned that GFP positive population contained a similar cellular composition in WT and *Zfp541* KO at P18, since *Zfp541* KO spermatocytes showed meiotic defects at late pachytene but before diplotene. Thus, a potential batch effect due to a bias in cellular population was minimized for the RNA-seq data. We clarified this in the revised manuscript (line 365-367).

As the reviewer suggested, now we have presented the expression profiles of DEGs by reanalyzing previous scRNA-seq data (GSE109033) of adult mouse testis (new Fig 5e). We added the related sentences in the revised manuscript (line 356-361).

Also, we clarified “most of the ZFP541-target genes that were upregulated in *Zfp541* KO were generally expressed in broad cell types rather than being germ-cell specific.” (line 405-406)

ChIP-seq using two novel ZFP541 antibodies predicted genomic binding sites in spermatocytes, but surprisingly, the simplest negative control – *Zfp541*KOs was not included – a very surprising oversight – and a very small proportion of the binding sites are associated with change in gene expression (and most changes in gene expression are not associated with a binding event). This emphasizes the two key concerns that the DEG analysis (bulk RNA-seq) was not done with an appropriate sample type and ChIP was not properly controlled.

We fully understand the reviewer’s point that *Zfp541* knockout cells should be used as a negative control for ChIP-Seq. However, it is difficult to perform ZFP541 ChIP-Seq with *Zfp541* KO cells due to the following reasons. In our ZFP541 ChIP-Seq analysis, $>1 \times 10^8$ fixed testicular cells isolated from juvenile mice were required for preparing libraries for ChIP-Seq. Since *Zfp541* KO males are infertile, we have to make homozygous mice by intercrossing heterozygotes. Therefore, it is hard to collect enough number of juvenile *Zfp541* KO testes at the same time. Thus, it is not realistic to do negative control ChIP-seq with *Zfp541* KO. (see also #17)

[Redacted]

As a control for antibody specificity, we validated our ChIP-seq data by using two different ZFP541 antibodies (supplementary Fig10a) and defined the common bound peaks as putative ZFP541-bound sites. The putative ZFP541-bound nearest genes were assigned regardless of the distance from the ZFP541-binding sites, of which 32.3 % (1912 nearest genes) resided around the TSS regions on the mouse genome (Fig. 6a). We appreciate that it is possible that not all of those nearest genes represented *bona fide* ZFP541 targets, but they solely represent just candidates of ZFP541 targets. Therefore, we tentatively defined those genes that showed differential expression in *Zfp541*KO vs WT as putative ZFP541 targets (Fig. 6d).

As an alternative analysis, we validated DEGs that putatively bound to ZFP541 was assessed by reanalyzing previous scRNA-seq data (GSE109033) of adult mouse testis (new Fig 5e, new Fig 6h). We toned down our interpretations by describing in the main text as follows (Line 423 -

429): “Since a subset of those putative DEGs act for general transcription processes, it is possible that some, if not all, of the DEGs are indirectly regulated by ZFP541. We do not exclude the possibility that DEGs putatively bound by ZFP541 are co-regulated by other transcription factors and/or epigenetic mechanisms. (see also #16).

In general, I found the data to be of very high quality, the experiments thoughtful and elegantly presented, but as noted above, the conclusions were not always supported by the results. Thus, there is significant room for improvement to better align the conclusions with the extent of results and provide meaningful molecular results in Figs. 5-6 to overcome technical problems. If the concerns can be addressed, then I believe this will represent a very valuable contribution to the field of spermatogenesis by revealing the requisite molecular players is extinguishing the meiotic program prior to spermiogenesis.

The following specific criticisms should also be addressed:

1. A number of typographical errors were noted throughout the manuscript which should be addressed upon revision.

These were revised.

2. Fig. 1A – Negative control ChIP with *Stra8* KOs should be shown.

Fig. 1A was done by the reanalysis of MEIOSIN and STRA8 ChIP-Seq data that we published previously (Ishiguro et al, Dev Cell 2020). Again, we understand the reviewer’s point that knockout cells should be used as a negative control for ChIP-Seq. However, it is difficult to perform in the short period for revision due to the following reasons. In our MEIOSIN ChIP-Seq analysis, $5-10 \times 10^7$ fixed pre-leptotene spermatocytes isolated from ~40 WT mice were required for one ChIP-Seq experiment. For this purpose, neonatal mice were daily injected with Win18,446 followed by RA injection to collect as many pre-leptotene cell population as possible. Since *Stra8* KO mice are infertile, we have to make homozygous mice by intercrossing heterozygotes, which takes long time to get enough number of neonatal *Stra8* KO males. Thus, it is not realistic to do negative control ChIP with *Stra8* KO.

Instead, we rephrased the sentence as “In **spermatocytes**, we identified *Zfp541* as one of the MEIOSIN/STRA8-bound genes during preleptotene”, since MEIOSIN and STRA8 ChIP-Seq data that were used for the reanalysis in this study were derived from pre-leptotene **spermatocytes**. (Line 108-119).

3. Fig. 1B – expression in *Stra8* KOs should be shown.

We examined *Zfp541* expression in *Stra8* KO by RT-qPCR. While *Zfp541* expression level was downregulated in *Meiosin* KO testis compared to P10 WT, it was comparable between P10 WT and *Stra8* KO testes. This is consistent with our previous observation that expression levels of some meiotic genes including *Zfp541* remained unaffected in *Stra8* KO (Ishiguro et al, Dev Cell 2020, RNA-seq data in Fig4H and Table S1). Previously, we pointed out that a subset of meiotic genes was more downregulated in *Meiosin* KO compared to *Stra8* KO, and preleptotene block was cytologically more severe in *Meiosin* KO than in *Stra8* KO. We assume that this is related to the relatively weaker phenotype of *Stra8* KO than *Meiosin* KO.

We presented a new set of data in Fig1b, and added related sentences (Line 114-118) .

4. Fig. 1E – this panel does not adequately assess expression of ZFP541 and show be expanded to show each stage of the cycle of the seminiferous epithelium with cell type markers and chromosome spreads (like in Fig. 2G) to confirm cell types. PNA co-labeling would be useful for identifying spermatids subtypes by stage to support claims of absence in elongating spermatids.

We newly presented immunostaining of seminiferous tubules by PNA co-labeling to assess the expression of ZFP541 (new supplementary Fig2a). ZFP541 signal started to appear faintly in early pachytene spermatocyte nuclei in stage I seminiferous tubules, and persisted in round spermatids in stage VII seminiferous tubules, but not in elongating spermatids. (line 137-140)

We confirmed that seminiferous tubules that contain PNA lectin positive spermatids were absent in *Zfp541* KO testis (new supplementary Fig4a-b). These data together with the HE staining of seminiferous tubules (Fig. 2d) indicate that post-meiotic spermatids were absent in *Zfp541* KO testis. We also added the related sentences in the main text (line 170-171).

5. Fig. 2G – the sex body in KOs hardly looks normal – need to confirm this with DNA FISH against the sex chromosomes.

We examined XY pairing in *Zfp541* +/- (n=52) and *Zfp541* KO (n=50) pachytene spermatocytes by immuno-FISH using Y painting probe and X point probe. The result indicates that X and Y chromosomes are apparently paired in most of the *Zfp541* KO pachytene spermatocytes that we examined (100%= 52/52 in the control, 98% = 49/50 in *Zfp541* KO, Chi-square test: $p = 0.1474$). This suggests that although heterochromatic XY body formation was impaired, X and Y chromosome pairing/synapsis was normal in *Zfp541* KO pachytene spermatocytes. We presented the immuno-FISH data in the new Fig2h and added the sentences in the main text (Line 195-198).

6. Fig. 2J – these data are insufficient to draw conclusions about MI – need metaphase spreads. MI spermatocyte can be identified by immunostaining of H3S10P +/- centromeric SYCP3. We did not find any tubule that contain spermatocytes with H3S10P +/- centromeric SYCP3 signals in *Zfp541* KO seminiferous tubules. Furthermore, we verified this observation by another immunostaining that could specifically detect MI spermatocytes (new Supplementary Figure 4c). MEIKIN, which we published before (Kim et al, Nature 2015), is a kinetochore protein and localizes to the kinetochore from late pachytene to metaphase I. Thus, MI spermatocyte can be identified by MEIKIN +/- centromeric SYCP3. In *Zfp541* KO, we did not find any tubule (stage XI-XII) that contained spermatocytes with MEIKIN +/- centromeric SYCP3 signals, except for the tubules (stage IX-X) that contained late pachytene spermatocytes with MEIKIN +/- axial SYCP3. Thus we concluded *Zfp541* KO spermatocytes failed to reach metaphase I stage. This is in contrast to the presence of stage XI-XII tubules that contain MI spermatocytes in *Kctd19* KO testis.

We added new data of immunostaining of the tubules by MEIKIN and SYCP3 (new supplementary Fig 4c). Because of absence of MI spermatocyte in *Zfp541* KO, we believe that metaphase spread experiment using *Zfp541* KO spermatocytes would not be informative.

7. Fig. 2K – TUNEL% of SYCP3+ vs. PNA+ would be much more informative than per tubule. Also, how were stages defined (authors claimed stages X-XI tubules, but no evidence provided)? Why are only <50% tubules effected?

As mentioned above, we performed immunostaining of seminiferous tubules using the markers PNA and SYCP3. We showed that the existence of the pachytene spermatocytes up to stage IX were comparable between WT and *Zfp541* KO, but PNA + post-meiotic spermatids were absent in *Zfp541* KO (new supplementary Fig.4a). Furthermore, we showed that TUNEL positive spermatocytes consequently appeared in the stage X tubule of *Zfp541* KO (new Fig.2 L). We revised the related sentences, and provided new data (new supplementary Fig.4a-b). Also, we replaced the WT (8w) panel of Fig2L with a new image, so that comparable stage X tubules were presented in WT and *Zfp541* KO.

We counted the tubules that have TUNEL+ SYCP3+ spermatocytes per total number of tubules. This analysis indicated that more than 50 % out of total tubules counted in *Zfp541* KO were yet to show TUNEL+ SYCP3+ spermatocytes.

8. Lines 184-185 – this sentence is not sufficiently supported by data.

We rephrased the sentence as ZFP541 is required for the completion of meiotic prophase and the transition to the meiotic division phase and spermiogenesis.

9. Fig. S2 – beautiful data!

We are happy to hear that. Thank you!

10. Fig. 3C-E should be expanded to consider stage of the epithelial cycle.

We newly presented immunostaining of seminiferous tubules by PNA co-labeling to assess the expression of KCTD19 (new supplementary Fig4, supplementary Fig7). KCTD19 signal started to appear in early pachytene spermatocyte nuclei in stage II-III seminiferous tubules, and persisted in round spermatids in stage VII seminiferous tubules, but not in elongating spermatids.

11. Fig. 4C – not sufficient to conclude nuclear localization.

Now we have presented the immunostaining of spread nuclei (new Fig3C), showing that KCTD19 appears in the nuclei of the spermatocytes from pachytene onward, and in round spermatids.

Since KCTD19 was hardly detected in the nuclei of the *Zfp541* KO seminiferous tubules despite the expression of KCTD19 in *Zfp541* KO, we rephrased the sentence as “nuclear localization of KCTD19 was impaired or KCTD19 protein was dispersed in the absence of ZFP541” (Line 281-282).

12. Fig. 4H – the bimodal distribution of MLH1 foci in KOs is very interesting – what is the explanation?

Although we still do not know the exact reason for this phenomenon, it is possible that meiotic recombination process may be compromised before MLH1 foci appear in some, if not all, *Kctd19* KO spermatocytes. We added this speculation in the main text (Line298-301).

13. Fig. 4K- This should also be done in *Zfp541* KOs.

Please see #6 above.

14. Fig. 4M – same comments as Fig. 2K.

We now have presented the immunostaining of seminiferous tubules in *Kctd19* KO with PNA co-labeling (new supplementary Fig4a). We confirmed that seminiferous tubules that contain PNA lectin positive cells were absent in *Kctd19* KO testis as well. This data together with the HE staining of seminiferous tubules (Fig. 4e) indicated that post-meiotic spermatids were absent in *Kctd19* KO testis. We added the related sentences in the main text (Line 287-289).

15. Fig. 5 – neither THY1 nor KIT are unique to spermatogonia, so these gene sets do not bin to Spermatogonia, as claimed. Suggest using single-cell datasets to provide a more robust and higher resolution separation of genes changed in the KO.

Our isolation protocol for THY1 and KIT spermatogonia has several steps to eliminate somatic cells prior to magnetic sorting (Maezawa, et al., 2018 NAR). Therefore, these fractions are largely specific to THY1 and KIT spermatogonia. We previously confirmed the high purity (> 95%) of these fractions (Maezawa, et al., 2018 NAR). To independently examine the expression profile of the DEGs in *Zfp541*KO, we analyzed the expression profiles of the DEGs in *Zfp541*KO using previously published scRNA-seq data of spermatogenic cells (Hermann et al., 2018) (Fig. 5e). This analysis indicated that those upregulated genes in *Zfp541* KO were overall found in early pseudotime, whereas the downregulated genes in *Zfp541* KO were found around mid to later stages of pseudotime. Among upregulated DEGs, 67.0 % (967 out of 1443 genes) were in the clusters representing spermatogonia. We placed the data in the new Fig. 5e, and added the related sentences in the main text (Line 358 - 361).

16. Fig. 6D genes (lines 317-320) – the mRNA expression profiles of the noted genes are highly complex across spermatogenic progression, and thus, their direct regulation by ZFP541 is not easy to establish – rather than oversimplifying, the authors should be more carefully present these as targets and show the relevant expression profiles.

We appreciate the reviewer’s thoughtful suggestions. We toned down our interpretations by describing in the main text as follows (Line 423 -429): “Since a subset of those putative DEGs act for general transcription processes, it is possible that some, if not all, of the DEGs are indirectly regulated by ZFP541. We do not exclude the possibility that DEGs putatively bound by ZFP541 are co-regulated by other transcription factors and/or epigenetic mechanisms.

17. Fig. 6 – validation/confirmation by ChIP-qPCR is lacking.

As mentioned above, in our ZFP541 ChIP-Seq analysis, $>1 \times 10^8$ fixed testicular cells isolated from juvenile mice are required for ChIP. Since *Zfp541* KO males are infertile, we have to make homozygous mice by intercrossing heterozygotes. Therefore, it is hard to collect enough number of juvenile *Zfp541* KO testes at the same time. Thus, it is not realistic to do negative control ChIP-qPCR with *Zfp541* KO spermatocytes. Also, in our negative control ChIP-seq using anti-GFP antibody in testis chromatin (Ishiguro et al, Dev Cell 2020), we confirmed that negative control antibody did not raise enrichments at those promoter regions. Thus, we assume that ZFP541 ChIP-seq peaks were not derived from non-specific binding of ZFP541 antibodies.

18. Fig. 7 – this model figure is highly speculative, especially when it comes to histone modification changes. This figure should be revised to only cover results supported by data at hand.

We revised the schematic model in the figure.

Reviewers' Comments:

Reviewer #1:

Remarks to the Author:

The diligence and effort of the authors in preparing this revision is greatly appreciated, as is the improvement in accuracy of timing in expression and phenotypes, which enhances knowledge gained and significance. The changes made fully resolve my previous concerns and suggestions.

Reviewer #2:

Remarks to the Author:

The addition of substantial new data and appropriate textual revisions to align results and conclusions in this revised manuscript from Horisawa-Takada is a significant improvement over the initial version. I appreciate the thoughtful and good-faith efforts to respond to quite a large number of criticisms and feel this will benefit the reader. This remains a compelling and important study and I have no further substantial criticisms.